



# The Sun's Role for Decadal Climate Predictability in the North Atlantic

Annika Drews[1,2], Wenjuan Huo[1], Katja Matthes[1], Kunihiko Kodera[3,4], Tim Kruschke[5]

[1]GEOMAR Helmholtz Centre for Ocean Research Kiel, 24118 Kiel, Germany
[2]SINTEF Ocean AS, 7010 Trondheim, Norway
[3]Meteorological Research Institute, Tsukuba, Ibaraki 305-0052, Japan
[4]RIKEN Nishina Center for Accelerator-Based Science, Wako, Saitama 351-0198, Japan
[5]SMHI - Swedish Meteorological and Hydrological Institute – Rossby Centre, 60176 Norrköping, Sweden

*Correspondence to*: Annika Drews (annika.drews@sintef.no)

**Abstract.** Despite several studies on decadal-scale solar influence on climate, a systematic detection of solar-induced signals at the surface and the Sun's contribution to decadal climate predictability is still missing. Here, we disentangle the solar-cycle-induced climate response from internal variability and from other external forcings such as greenhouse gases. We utilize two 10-member ensemble simulations with a state-of-the-art chemistry climate model, to date a unique data set in
chemistry climate modelling. We quantify the potential predictability related to the solar cycle and demonstrate that the detectability of the solar influence on surface climate depends on the magnitude of the solar cycle. Further, we show that a strong solar cycle forcing organizes and synchronizes the decadal-scale component of the North Atlantic Oscillation, the dominant mode of climate variability in the North Atlantic region.

## 20   1 Introduction

Since the middle of the last century long-term changes in global climate have been dominated by anthropogenic greenhouse gas emissions. Nevertheless, natural forcings play an important, but poorly quantified role in past and present climate, especially on regional scales. Solar forcing variability has been suggested to affect regional climate variability (Gray et al., 2010) and to synchronize internal variability modes such as the North Atlantic Oscillation (NAO) (Thiéblemont et al., 2015).
Consequently, solar variability may offer a source of decadal predictability for regional climate due to its periodicity (Dunstone et al., 2016; Kushnir et al., 2019).

In recent years, comprehensive (decadal or near-term) climate prediction efforts have been made to provide a skilful and reliable forecast of the actual evolution of both externally forced and internally generated components of the climate system. These prediction systems show forecast skill for several years (Bellucci et al., 2015; Yeager and Robson, 2017) beyond the
externally forced climate response (Smith et al., 2019). Yet, a comprehensive understanding of predictability of the coupled climate system as well as the interaction of different predictability drivers is missing.

It is challenging to separate the 11-year solar cycle surface signal from internal climate variability in observations because the solar signal is small compared to internal variability and the number of solar cycles is limited. Investigating its



detectability, a recent study (Chiodo et al., 2019) questioned the statistical reliability of the previously widely accepted solar
cycle influence on North Atlantic climate projecting onto the NAO (Gray et al., 2013, 2016; Kodera, 2003; Kodera et al.,
2016; Kodera and Kuroda, 2002; Matthes et al., 2006; Thiéblemont et al., 2015). The present manuscript intends to partly
rebut the conclusions of this study and to provide new robust evidence of solar influence on North Atlantic climate. We
utilize a unique set of two 10-member ensemble simulations of a state-of-the-art coupled chemistry climate model to isolate
and decipher the 11-year solar cycle's footprint in North Atlantic surface climate variability and to quantify the contribution
of the solar cycle to regional decadal potential predictability relative to other external forcings and internal variability during
Northern Hemisphere winter. These simulations include a realistic solar forcing dataset recommended for CMIP6 (including
solar radiative and particle forcing), a well-resolved shortwave radiation scheme, a comprehensive module for middle
atmosphere chemistry modelling as well as an interactive ocean to capture internal climate variability adequately.

## 2 Potential predictability associated with the solar cycle

The chemistry-climate model in use is CESM1(WACCM) (Marsh et al., 2013). One ensemble includes the full CMIP6 solar
forcing (FULL), while the other only considers the low-frequency (time scales longer than ~30 years) changes of solar
irradiance (LOWFREQ) (Fig. S1). All simulations have been integrated over the historical period 1850-2014. We estimate
the potential predictability variance fraction (ppvf) related to the 11-year solar cycle as well as to all other external forcings
(including the low-frequency component of solar variability) for decadal (8-year running mean) variations of winter (DJF)
surface air temperatures (see "Methods"). 8-year running means are chosen as these are a typical target of actual decadal
prediction efforts (Goddard et al., 2013). The ppvf describes how much of the total decadal variance in our FULL ensemble
is explained by the respective forcing(s). The ratio of variance that cannot be associated with any external forcing is
considered as internal variability. The extratropical North Atlantic is a hotspot of solar cycle influence on climate
predictability (Fig. 1a) where up to 25% of the decadal variability of winter surface air temperatures are explained by the
solar cycle. At the same time, this region shows low potential predictability due to other (low-frequency) external forcings
(Fig. 1b) and large internal variability (Fig. 1c).



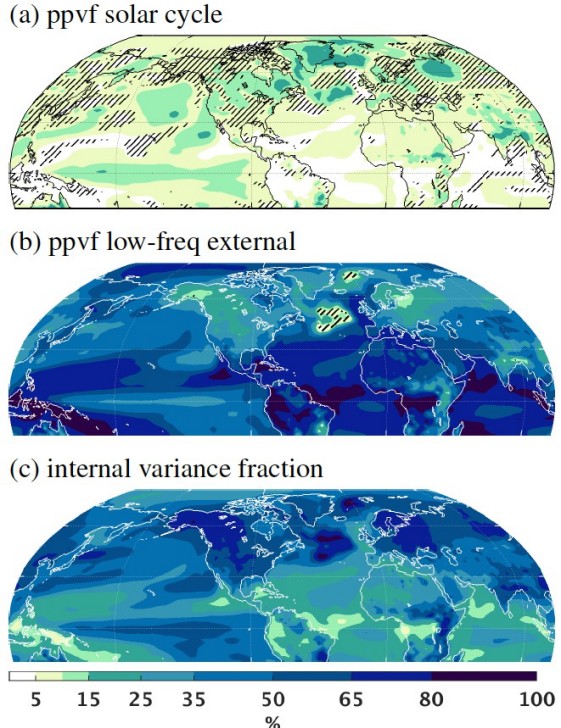

**Figure 1: Decadal potential predictability due to the solar cycle, other external forcings and internal climate variability.** Potential predictability variance fraction (ppvf; explained variance) with respect to the DJF 8-year averaged surface air temperature associated with (a) the 11-year solar cycle, (b) all other external forcings (anthropogenic forcings, volcanic aerosols, and solar-induced low-frequency variability), and (c) remaining variance fraction due to internal climate variability. Statistically insignificant regions ($p>0.05$) in Figs. 1a and b are hatched. Fig. 1c exhibits masked areas solely over a few grid points in the North Atlantic because the "ppvf low-frequency external" for 2m-temperature is significant almost globally. See "Methods" section for more details on the ppvf calculation and statistical significance estimates.





The range of externally and internally generated variance fraction of our large CESM ensemble is well within the range of other high-top CMIP5 models (Fig. S2), which do not distinguish between solar cycle and other external forcings however. Also, our results agree qualitatively well with previous studies showing (i) low potential predictability due to external forcings and large internal variability over the extratropical North Atlantic (Boer et al., 2013), and (ii) statistically significant

surface signals associated with the solar cycle (Gray et al., 2010, 2013; Kodera, 2003; Thiéblemont et al., 2015). This is further supported by Figure S3 when comparing the "skill" (correlation with observations) of FULL and LOWFREQ for the North Atlantic region. Consequently, solar variability and an adequate representation of its impact on climate is key to exploit the solar-induced potential predictability for decadal climate predictions.

## 3 The top-down mechanism depends on solar cycle amplitude

The presence of large internal variability during winter complicates the detection of the small solar-cycle-forced signal in the short observational record or even in long climate model simulations. We successfully isolate the 11-year solar cycle response based on two 10-member ensemble simulations - to date a unique data set in chemistry climate modelling. We separate it from other external forcings by subtracting the LOWFREQ ensemble mean from the FULL ensemble, and from internal variability by averaging over the individual ensemble members (see "Methods").

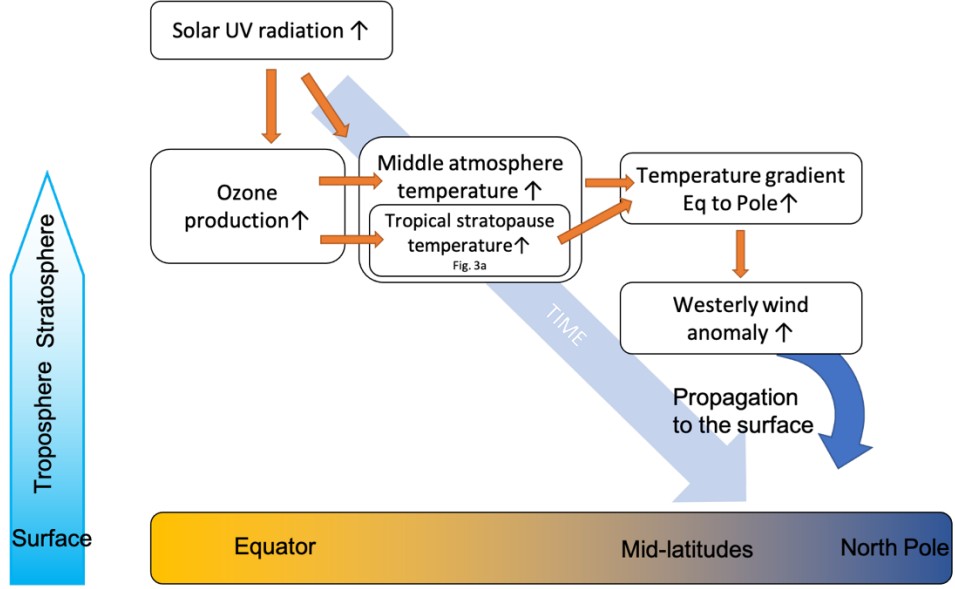


**Figure 2: Top-down mechanism.** Enhanced solar UV radiation during solar maxima enhances heating of the middle atmosphere and ozone production, which in turn enhances heating in the tropical stratopause (Gray et al., 2010; Haigh, 1994). As a result of this anomalous

heating in the tropical upper stratosphere, an increased meridional temperature gradient (Fig. S4) initiates a westerly wind anomaly in the upper mid-latitude stratopause region (Fig. S5), which through wave-mean flow interaction changes the propagation properties for planetary waves and transports the early winter signal to the lower stratosphere, troposphere and to the surface in late winter (Fig. S5).



The top-down mechanism of solar influence on surface climate is depicted in Figure 2 and we refer to the plethora of
literature (e.g., Gray et al., 2010; Haigh, 1994; Kodera and Kuroda, 2002; Matthes et al., 2006). Our ensemble supports this
general and widely accepted concept. The tropical stratopause temperature anomaly varies in phase with the 11-year solar
cycle (Fig. 3a). As the solar cycle amplitude shifts from a "weak epoch" into a "strong epoch" (see "Methods" for distinction
between weak and strong epoch), the variance of the solar-induced temperature changes compared to the magnitude of
internal variability increases from 31% to 69%, and the correlation with the F10.7 index from $r_{T\_weak}=0.55$ to $r_{T\_strong}=0.72$.
The ensemble mean zonal wind gets more organized and in phase with the solar forcing during the strong epoch (Fig. 3b):
The correlation between the solar index and the solar-induced ensemble mean wind changes is close to zero ($r_{U\_weak}=-0.12$)
during the weak epoch but rises to $r_{U\_strong}=0.36$ in the strong epoch. However, its amplitude does not differ much between
the weak and the strong epoch (its variance is 22% (weak epoch) and 23% (strong epoch) compared to internal variability).
During the weak epoch, the weak direct solar signal in tropical stratopause temperatures (Fig. S4) is not strong enough to
maintain the stratopause circulation longer in a radiatively controlled state, the anomalous westerly winds shift poleward
already in the middle of November and are then controlled by "non-forced" polar dynamical processes (Fig. S5, S6) (Kodera
et al., 2016; Kodera and Kuroda, 2002). This fast transition from a radiatively controlled to a dynamically controlled state
warms up the entire polar stratosphere through a modulation of the Brewer-Dobson circulation (Fig. S6). In contrast, in the
strong epoch, the radiatively controlled state switches to a dynamically controlled state as late as in February: We find the
"typical" downward propagation of zonal wind anomalies in later winter and a synchronization of the ensemble members
(Fig. S5). This shows that the response to the solar cycle is highly non-linear and not necessarily proportional to the forcing.
Therefore, the following analyses concentrate on the strong epoch only where a synchronization takes place.

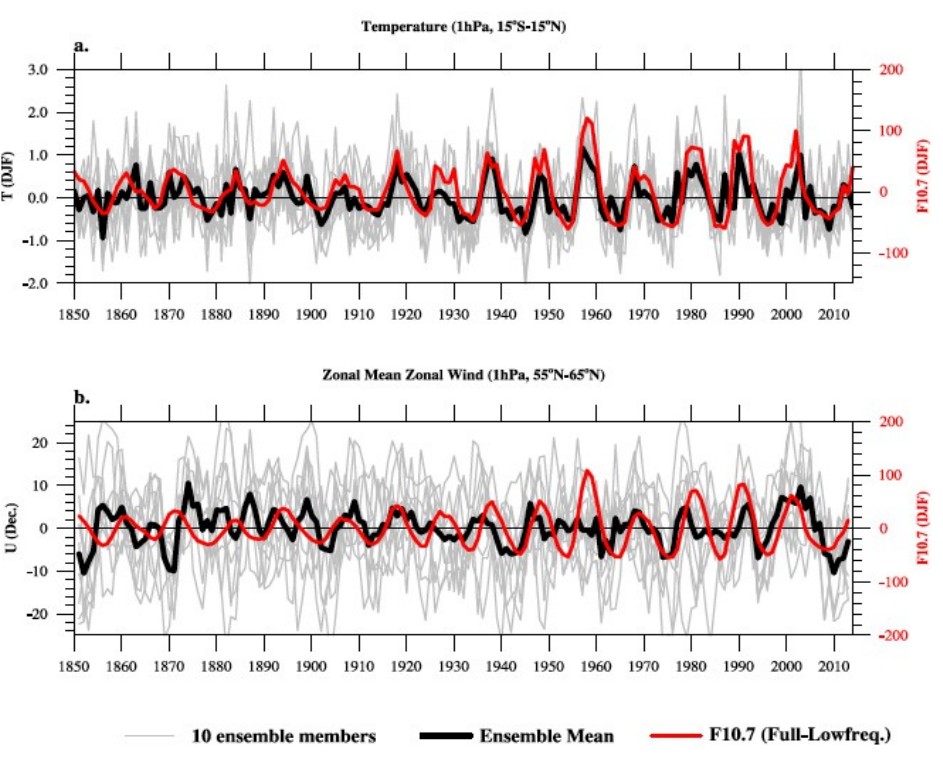






**Fig. 3. Direct solar cycle signal in the upper stratosphere.** a) Time series of the tropical stratopause temperature (1hPa, 15°S-15°N) averaged over the winter season (DJF) for the individual ensemble members (grey thin lines), ensemble mean (black thick line) and the solar cycle index F10.7 (red line). b) Same as a), but for December zonal mean zonal wind averaged over 55-65°N, 1hPa, smoothed with a 3-year running mean. To isolate 11-year solar cycle effects, differences between the FULL and the LOWFREQ ensemble simulations have

been calculated. The grey lines include internal variability and the 11-year solar signal, their spread represents the internal variability, and the ensemble mean (thick black line) represents the 11-year solar cycle signal. The 11-year solar cycle has been isolated in a similar way from the original solar forcing time series by subtracting the low-frequency part of the solar forcing (Fig. S1).

## 4 Solar-induced surface signals and synchronization of the NAO with the solar cycle

Consistent with the zonal mean zonal wind anomalies (Fig. S5) a clear and statistically significant surface response appears in sea level pressure (SLP) in February featuring higher pressure anomalies in the mid-latitudes and lower pressure anomalies over the North Pole with a minimum over Scandinavia and the Norwegian Sea representing a negative Annular Mode pattern during the strong epoch (Fig. 4a). Consistent with the annular SLP and wind signals a statistically significant response in sea surface temperatures (SSTs) in the North Atlantic appears (Fig. 4b). The model response is strongest at the

time where the strongest zonal mean zonal wind signal extends down to the surface, propagating poleward-downward from the stratosphere (Fig. S5).

The solar signal in SLP resembles the tri-polar NAO pattern (Visbeck et al., 2003), but it is shifted towards Europe with centers over Scandinavia and the Mediterranean Sea (and a secondary maximum towards North America) as compared to the NAO, which has action centers close to Iceland and the Azores (Fig. 4a). To further investigate the role of the solar cycle for

decadal climate variability in the North Atlantic region, we focus on the NAO-like solar signal in the following.

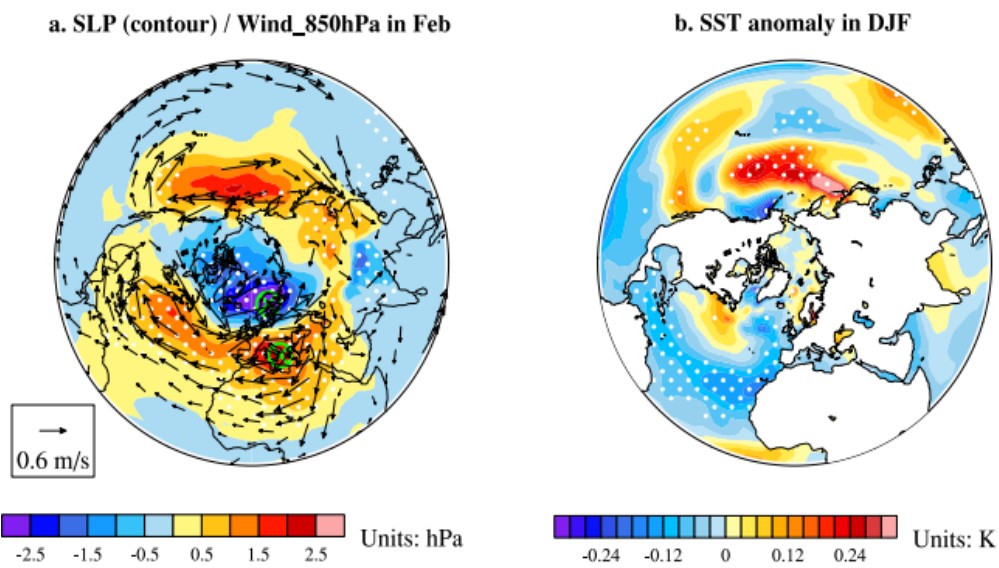



**Figure 4: Solar surface signal in NH winter at lag 0 during the strong epoch.** Composite differences between the solar maximum and minimum for a) SLP (contours) and wind at 850hPa (vectors) in February. Only those vectors where the zonal mean wind component is significant at the 90% level are shown. The two green circles indicate the stations used for the NAO index calculation (see Method section for more details). b) same as a) but for SST (contours) averaged over the winter season (DJF). White dots indicate 90% statistically significant regions based on a 1000-fold bootstrapping test.

The NAO-like mode (green markers in Fig. 4a; see "Methods") exhibits typical quasi-decadal internally generated oscillations during NH winter (cf. blue line in Fig. S7). The solar cycle synchronizes the phase of the NAO-like index in February during the strong epoch (Fig. 5a), while the internally generated decadal-scale NAO has no clear phase relationship with the solar cycle (Fig. 5b). The relationship between the solar cycle and the model ensemble mean NAO index is stable only during the strong epoch when the solar cycle forcing is strong. Their running correlation for all overlapping 45-year windows is fluctuating in the earlier years but begins to rise in the 1920's both for the model and observations (Fig. 5c). In the second half of the 20th century, the correlations reach statistical significances of 90% in the model and in observations with a lag of two years (Gray et al., 2013; Thiéblemont et al., 2015) (see "Discussion"). A comparison of the solar signal with internal variability reveals that the solar signal in SLP over Europe is approximately 18% the magnitude of internal variability during the strong epoch. The solar cycle signal is small in magnitude but manifests itself as an organization and synchronization of internal variability as shown by the cross-correlations (Fig. 5a) and the increasing running correlations (Fig. 5c), which is in agreement with earlier studies (Otterå et al., 2010; Scaife et al., 2013; Thiéblemont et al., 2015). We here show that this "organization" depends on the solar cycle amplitude and it is large enough for a potential predictability variance fraction (ppvf) of up to 25% in the North Atlantic region (Fig. 1a).





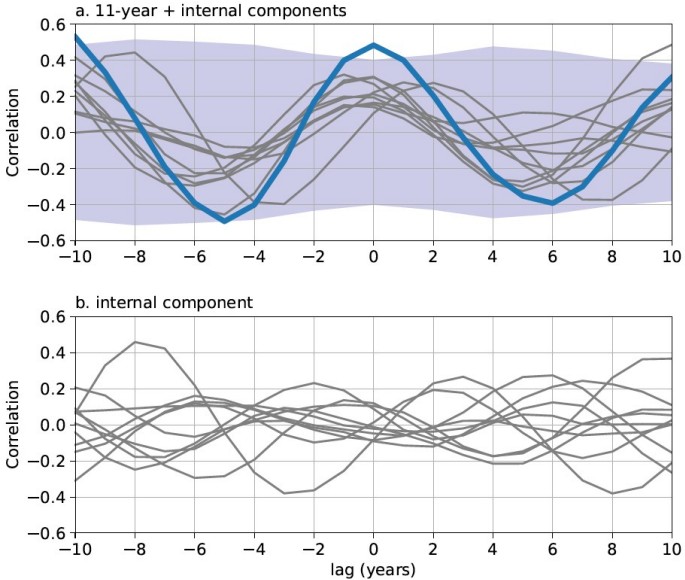

155

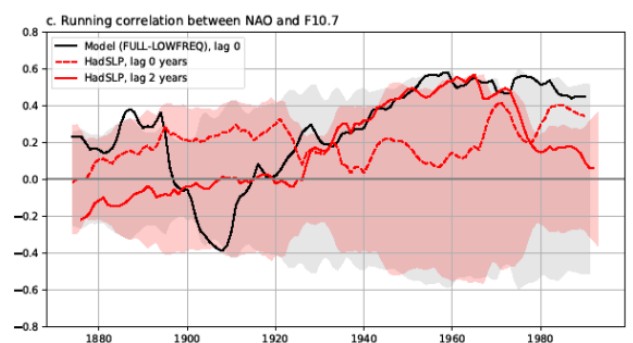

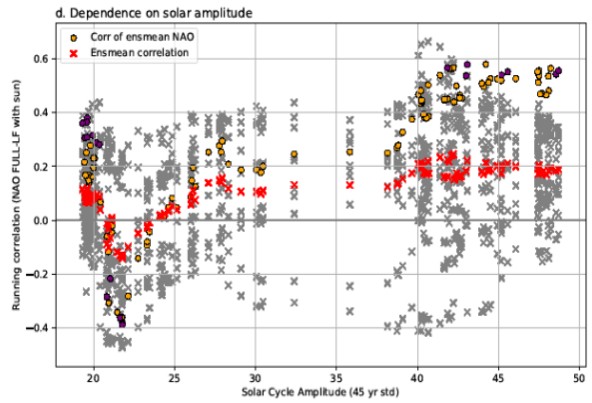





**Figure 5: Synchronization of internal variability modes in the Atlantic with the solar cycle.** Cross correlation between the wintertime (DJF) solar index (F10.7cm solar flux) and the February NAO-like station-based indices for a) 11-year + internal components (FULL minus ensemble mean of LOWFREQ) and b) internal component (FULL minus ensemble mean of FULL) during the strong epoch (gray thin lines: individual members; blue line: ensemble mean). The 90% significance level is shown as shaded envelope (light blue). c) 45-year running correlation of NAO indices in February from the model (black lines) and observation (red solid (lag 2 years) and red dashed (lag 0 years) lines) with the F10.7 index. The year on the x-axis denotes the central year of the window (in case of the lagged time series it is w.r.t. the NAO index). All indices are station-based and smoothed with a 3-year running mean prior to calculating the correlation. The 90% significance level is shown as shaded envelopes, for model (grey) and observations (light red) separately. d) Scatter plot of February NAO-sun running correlations (FULL-LOWFREQ, smoothed with 3-year running mean, 45-year windows (similar to Fig. 4c) against solar cycle amplitude (standard deviation of smoothed F10.7cm time series, same 45-year windows). Red crosses are the ensemble mean of correlations of single ensemble members with the solar index. Dots are the correlation of the ensemble mean NAO with the solar index, purple dots are correlations that exceed the 90% significance level calculated with a random-phase test.

Note that the observations only provide one realization, which includes all external forcings as well as internal variability. We show that a significant solar signal in the North Atlantic region is generated by the 11-year solar cycle in our 10 ensemble simulations during the strong epoch. The alignment of the surface pressure pattern with the solar cycle after the 1920's can also nicely be seen in the NAO-like time series (Fig. S7), which agrees with a number of earlier studies (Dima et al., 2005; Gray et al., 2016; Ma et al., 2018). A weak imprint can even be seen in the EOF-based NAO index (Fig. S7).

The dependence of the correlation strength on the solar cycle amplitude can be depicted from Fig. 5d: while for low solar cycle amplitudes correlations are scattered around zero, they are larger and positive for enhanced solar cycle amplitudes. Note that we do not expect larger correlations since the NAO index represents near-surface signals and as discussed above the solar surface signal is small. Hence, even though correlations as in Fig. 5c are similar to what has been shown in a recent study (Chiodo et al., 2019), our conclusions are very different.

## 5 Discussion

Recent analyses of the solar influence on the North Atlantic revealed insignificant responses (Chiodo et al., 2019) whereas numerous previous studies have found statistically significant solar cycle signals in surface climate in observations and climate models (Gray et al., 2010, 2013, 2016; Kodera, 2003; Kodera et al., 2016; Matthes et al., 2006) and even proposed a synchronization of the NAO by the 11-year solar cycle through the "top-down" stratospheric mechanism (Thiéblemont et al., 2015).

Here, we systematically detected solar-induced atmospheric and surface signals and separated them from internal variability as well as other external forcings based on two 10-member ensembles. For the first time, it was possible to overcome the problems in observations where a separation of solar-forced variability and a comparison to other external forcings and internal variability is impossible.

We demonstrate that the strength of the solar surface signal depends on the amplitude of the solar cycle. A stronger signal is detected during the "strong solar cycle epoch" and explains the limited detectability of observed surface signals before the 1930's as well as discrepancies in earlier studies. We show that a stronger solar cycle signal induces a surface response that resembles the NAO. The NAO in turn is organized and synchronized by the solar cycle. This is because the solar cycle





enforces the NAO phase if the solar forcing is strong enough. Our study confirms previous studies (Misios and Schmidt, 2012; Thiéblemont et al., 2015) which used the strong epoch for their solar forcing.

While we find statistically significant ensemble mean solar-cycle-induced surface signals in February during the strong epoch which are consistent with the top-down propagation from stratospheric signals, a controversial recent study (Chiodo et al., 2019) making use of combined DJF surface pressure pattern could not detect significant surface signals in their SOL

simulation. Our results suggest that it might be necessary to analyze monthly fields to capture a highly monthly varying signal. Also, although the base model is quite similar, there are a couple of important differences. Whereas we use two 10-member ensemble simulations of the historical period (2x10x165years=2x1650years; for the strong epoch this means: 2x10x83=2x830years) with the CMIP6 solar forcing (radiative and particle forcing) and the anthropogenic signal included, the other study uses two 500-year integrations with perpetual anthropogenic conditions representative for the year 2000 and

an idealized solar forcing which repeats the last four solar cycles (cycle 20-23) and does not include the low-frequency component over the historical period. For the analysis of sea level pressure or zonal mean winds relatively short 100 or 30-year periods are considered, whereas we have 10 members of 83 years each for the strong period. We isolate the 11-year solar cycle signal by subtracting the LOWFREQ ensemble mean from the FULL ensemble, whereas the other study compares the simulation which includes the idealized 11-year solar cycle (SOL) with a simulation that does not include any

solar forcing (NOSOL).

We would like to note that there are still some discrepancies between our simulations and observations, such as the different timing and location of maximum responses. We do not find a lagged NAO response in our simulations and the synchronization can only be found in February when the solar-induced zonal mean wind reaches the surface, while the largest response in observations appears at a lag of two years (Fig. S9). Possible reasons for these discrepancies are (i) that

the observational record is only one ensemble member that includes all internal variability and responses to all external forcings, (ii) that the model feedback from the ocean is insufficient and (iii) the influence of the solar cycle is highly non-linear which is very difficult to grasp.

The importance of the periodic solar cycle is particularly interesting for decadal climate predictions. Our results indicate that the solar cycle has a significant contribution to the potential predictability of up to 20-25% in the North Atlantic during

winter, where the climate response to external forcings such as greenhouse gases is comparably low. This offers the opportunity to increase decadal prediction skills in particular over Europe. This is in line with a very recent study which found exactly this region to be particularly sensitive to natural external forcings such as volcanic aerosols and solar activity when comparing decadal prediction systems from CMIP5 and CMIP6 (Borchert et al., 2021). The key for climate models to exploit this potential originating in solar forcing is - besides a sufficiently large ensemble (see Fig. S8) - the incorporation of

necessary pre-requisites to simulate the effects of the solar cycle: a realistic, spectrally resolved solar forcing dataset, either interactive chemistry or at least an ozone forcing including an estimate of the solar cycle impact and a well resolved shortwave radiation scheme to account for the solar UV changes.



## Methods

**Model description.** We use the fourth version of the Whole Atmosphere Community Climate Model (WACCM4) (Marsh et al., 2013) which is part of the Community Earth System Model suite (CESM) version 1.0.6 (Hurrell et al., 2013). CESM1(WACCM) is an extension of the Community Atmospheric Model (CAM4) (Kinnison et al., 2007; Neale et al., 2013). It covers the atmosphere from the surface to the lower thermosphere ($4.5x10^{-6}$ hPa; approx.145~km) and is considered a "high-top" model. CESM1(WACCM) has a horizontal resolution of 1.9°x2.5° and 66 vertical levels. It includes a middle atmosphere chemistry module which is based on the Model for Ozone and Related Chemical Tracers (MOZART3) (Garcia et al., 2014), comprising a total number of 59 species and 217 gas-phase chemical reactions (including all members of the $O_X$, $NO_X$, $HO_X$ chemical groups). For the presented simulations, we implemented a set of model improvements (Garcia et al., 2014, 2017; Smith et al., 2014). These include an increased turbulent Prandtl number yielding an increased diffusion coefficient as well as modifications related to gravity waves: (i) the orographic gravity wave drag does not depend on the land fraction of the gridbox anymore and (ii) the ratio of energy from dissipating gravity waves that is transformed into heat has been reduced to 30%. Since this model version is not able to internally generate a Quasi-Biennial Oscillation (QBO), we relax stratospheric equatorial winds towards an idealized QBO with a fixed 28-months period (Matthes et al., 2010). The POP ocean module has a tripolar horizontal grid of 1°x1° and 60 depth levels.

**Experimental design.** The two 10-member ensemble simulations, FULL and LOWFREQ, cover 165 years each from 1850 through 2014, follow the CMIP5-historical recommendations for all external forcings from 1850-2004 and continue with the RCP4.5-scenario afterwards. Only for the solar forcing the improved CMIP6 dataset has been used (Matthes et al., 2017). The FULL ensemble experiences the complete solar variability from the CMIP6 dataset with total and spectrally resolved solar irradiances as well as energetic particle effects. The LOWFREQ ensemble only includes the low frequency part of solar variability by low-pass filtering the solar forcing with a 33-year running mean (Fig. S1). The individual ensemble members of FULL and LOWFREQ have been each initialized from different climate states of a multi-centennial pre-industrial control simulation.

**Observations.** The NOAA Global Surface Temperature (NOAAGlobalTemp V4.0.1) including The Extended Reconstructed Sea Surface Temperature (ERSST) dataset (version 5) (Huang et al., 2017) and land surface air temperature data from the Global Historical Climatology Network-Monthly dataset (Menne et al., 2018) is used to calculate correlations with the model simulations. HadSLP2r data (Allan and Ansell, 2006) is used to calculate the observed NAO index as well as composites. All these data is provided by the NOAA/OAR/ESRL PSL, Boulder, Colorado, USA, from their Web sites at https://psl.noaa.gov/data/gridded/data.noaaglobaltemp.html and https://psl.noaa.gov/gcos_wgsp/Gridded/data.hadslp2.html .

**Disentangling internal and external variability.** The single members in one ensemble (FULL or LOWFREQ) share the same externally forced signals but have different internal variability. To isolate the internal variability from external forcings, the ensemble average of the respective ensemble (FULL or LOWFREQ) has to be subtracted from the individual members. The ensemble mean (average over all individual members) represents the combined effects of all external forcings.





External forcings in the LOWFREQ ensemble comprise low-frequency solar variability as well as changes in greenhouse gases, while in the FULL ensemble additionally the effect of the 11-year solar cycle is included. In order to isolate the solar cycle from other external forcings, the ensemble mean of the LOWFREQ ensemble has to be subtracted from the ensemble mean of the FULL ensemble.

**Definition of weak epoch and strong epoch.** Two periods (weak and strong epoch) with different solar cycle amplitudes were analyzed separately. Considering the continuous integration to capture the climate evolution with changing solar and anthropogenic forcing similar to observations, we defined the weak epoch from 1850 to 1931 and the strong epoch from 1932 to 2014. Important to note is, that each epoch includes 7 entire solar cycles but with different solar cycle amplitudes. The solar cycle amplitudes in the weak and strong epochs defined by the difference between the smoothed F10.7cm

maximum and minimum (red and blue dots in Fig. S1) of each cycle are shown in Table. S1, as well as the standard deviation of the smoothed F10.7cm in each cycle. A different sorting of the weak and strong solar cycle amplitudes was tested since cycle 15 is stronger than others in the weak epoch and cycle 20 is weaker than others in the strong epoch, but it was rejected since only one solar cycle (cycle 15) in the weak epoch showed slightly enhanced amplitude and it was checked that this does not affect our results.

**Decadal potential predictability.** The potential predictability variance fraction (ppvf) (Boer, 2004) is used to quantify the fraction of the decadal variability (8-year running mean) (i) forced by the 11-year solar cycle, (ii) forced by all other external forcings (including the low-frequency component of solar variability), and (iii) due to internal variability. First, we calculate the variance of each of these components (disentangling solar cycle, external and internal components as described above), then divide it by the total variance to obtain the ppvf of each component. An 8-year running mean was applied to surface

temperatures before calculating the variance in order to isolate decadal variability. Statistically insignificant regions ($p > 0.05$) in Figs. 1a and b are estimated by analyzing the ppvf attributed to the solar cycle and the low-frequency external components in comparison to the internal variance fractions by means of a Fisher's f-test in line with earlier studies (Boer, 2004). Calculations in Fig. 1a take into account the reduction of effective sample size due to the application of a low-pass filter.

**Composites, correlations, and statistical significances.** Solar cycle-based composites are used to identify solar signals. The

monthly time series of the F10.7 cm radio flux in solar flux units (1 sfu = $10^{-22}$ Wm$^{-2}$Hz$^{-1}$) is smoothed with a 3-year running mean (orange line) to determine local solar maxima and minima in January as representative for boreal winter (Fig. S1). Three years around each maximum and minimum are selected (dots in Fig. S1). Solar composites are then calculated as differences between averages of all solar maximum years and all solar minimum years. A 1000-fold bootstrapping test with replacement (Diaconis and Efron, 1983) is used to estimate the 90% statistical significances of the ensemble mean

composites compared to the individual ensemble members. Cross correlations and running correlations are calculated with a 45-year window to investigate the relationship between the 11-year solar cycle forced signal and natural internal decadal variability. The significance test for correlations was calculated based on 10,000 random time series with random phases (Ebisuzaki, 1997).



**NAO index definition.** The model NAO-like index was created by selecting locations close to the centers of the minimum
and maximum of the solar-induced SLP composites in February (Fig. 4a, green markers, 65.37N/22.5E, and 38.84N/17.5E).
Note that this NAO definition is comparable to a landmark study (Exner, 1913) which described the spatial structure of the
NAO first (Hurrell and Deser, 2015). For observations, the HadSLP2r dataset was used with stations in Reykjavik/Iceland,
and Lisbon/Portugal. First the February SLP at each location was normalized (all-time mean subtracted and then divided by
the standard deviation over the time series), then the difference between the Northern and the Southern station was
calculated. For the model data, this was done for every member in the FULL and LOWFREQ ensemble, before calculating
the two ensemble means. Finally, the LOWFREQ ensemble mean was subtracted from the FULL ensemble mean, to
investigate the effects of the solar cycle only. The running correlations in Fig. 5c were calculated by (i) applying a 3-year
running mean to the NAO-like time series and the F10.7cm time series of DJF averages, (ii) choosing the same 45-year
windows for both time series (or shifted by 2 years for the lagged correlations) and correlating them, moving forward one
year at a time. The year at the x-axis denotes the central year of the 45-year NAO-like time series window.

**Code availability**

The source code of the Community Earth System Model version 1.0 (CESM 1.0.6) used in this study is publicly distributed
and can be obtained after registration at http://www.cesm.ucar.edu/models/cesm1.0/. Codes to reproduce all figures are
stored in a git repository at https://git.geomar.de and are available from the corresponding author upon reasonable request.

**Data Availability**

All raw CESM model output is archived at the DKRZ. Post processed observational datasets and CESM model output is
archived in a Git LFS repository at https://git.geomar.de and are available from the corresponding author upon reasonable
request.

**Author contribution**

A.D. did the analysis and produced Figs. 2 and 5 as well as Figs. S1 and S7 in the supplementary material, W.H. did the
analysis and produced Figs. 3 and 4, Figs. S3-S6 as well as Fig. S8 and S9 in the supplementary material, T.K. produced Fig.
1 and Fig. S2, all in correspondence with K.M., who initiated the study. T.K. initiated and performed the model simulations.
K.K. assisted with the interpretation of the results. A.D. and K.M. wrote the final version of the manuscript with input from
all co-authors.

**Competing interests**

The authors declare that they have no competing interests.

**Acknowledgments**

We would like to thank a number of colleagues for continuous discussions about the solar influence on climate during the
last years, in particular Bernd Funke as well as Lesley Gray, Joanna Haigh, Adam Scaife, and Hauke Schmidt. We are
grateful for continued computing support and resources provided by the Deutsche Klimarechenzentrum (DKRZ) in



Hamburg, Germany where all simulations have been performed. We thank J. Kjellsson for help with downloading the CMIP5 high-top model data. This work has been conducted in the frame of the WCRP/SPARC SOLARIS-HEPPA activity.

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
