# Peer review of "The Sun's Role for Decadal Climate Predictability in the North Atlantic"

_Atmospheric Chemistry and Physics, 2021_

## Referee Comment (RC1)

Review of "The Sun's Role for Decadal Climate Predictability in the North Atlantic" submitted by Drews et al. for publication in ACP

The authors interpret two ten-member ensembles of CMIP-type "historical" simulations with respect to solar cycle signals in the boreal winter and in particular the North Atlantic surface climate. I think that simulations and their analysis are very appropriate for the purpose, writing and figures are in general very clear, such that I expect the manuscript to become publishable after consideration of a few issues that I will elaborate below. Overall I qualify the, in my view, necessary revisions as major although they may require relatively little effort. My issues with the current form of the manuscript are largely related not to the results themselves, but to their interpretation, discussion and framing.

The authors write that they intend "to partly rebut the conclusions" of a study by Chiodo et al. (2019) which speaks in its title of an "insignificant influence of the 11-year solar cycle on the North Atlantic Oscillation". My understanding is that the authors don't question the observational part of the analysis of that paper, but the analysis of observations which was done for DJF means, while in this manuscript, a surface signal significant at the 90% level is detected for February. The authors conclude "that it might be necessary to analyze monthly fields to capture a highly monthly varying signal". Why speculate? I suggest to either analyze the Chiodo et al. model output for monthly signals and/or the new simulations for DJF signals. The authors mention further differences, as the missing low-frequency solar variation in the Chiodo et al. simulations. But why should this matter given that the authors make an effort to exclude this part of the solar signals from their analysis. More generally, it would be appropriate to discuss the agreement or disagreement of this study's result with those of other papers more carefully. It seems the authors see their study in agreement of other papers they cite, e.g. in the Discussion "(Gray et al., 2010, 2013, 2016; Kodera, 2003; Kodera et al., 2016; Matthes et al., 2006)", which they oppose to Chiodo et al.. However, several of those studies actually discuss North Atlantic surface signals only for DJF, so it seems to me that the actual results of this paper (no DJF signal) are rather similar to those of Chiodo et al., and it is mostly the framing where it differs.

My second major point is that I think the authors do not adequately compare their model results to the available observation-based datasets. In the Discussion they mention that they "do not find a lagged NAO response… while the largest response in the observations appear at a lag of two years." However, in the results section, when discussing Fig. 5c this is much less clearly presented. They write, e.g., that "running correlation … begins to rise in the 1920's both for the model and observations." But this is not at all the case for the observations analyzed for lag zero. The following sentence is probably unintentionally unclear about "lag of two years" relating to model or observations or both. Why not, to compare apples and apples, also include model results for lag +2 in Fig. 5c? Moreover, it would be useful to identify also for other analyses if the observations are in the range of results provided by the individual ensembles, even if, unfortunately, one can neither conclude with certainty from such an agreement that the model is correct, nor that the observations provide a typical signal.

Already in the abstract, the authors claim to "show that a strong solar cycle forcing organizes and synchronizes the decadal-scale component of the North Atlantic Oscillation". They use this formulation a few more times. I claim they only show that this is true in their model

reality. Of course, this is very useful, and the same mechanisms could also act in reality, but we can't be sure. This in particular the case because of the involved non-linearities in the system, mentioned several times by the authors, and the apparently very different response to only slightly different forcings (compare weak and strong forcing epochs).

My last general point concerns the interpretation of the results with respect to decadal predictability. If this is supposed to be the main point, as the title suggests, I think this needs more careful and enhanced discussion. For instance, the authors claim that they use 8-year averaging because this is "a typical target of actual decadal prediction efforts". However, in the reference they mention for this (Goddard et al., 2013) it is said that their choice of 1, 4, and 8 years may seem arbitrary, but was chosen to illustrate the effects of different temporal averaging. Many actual decadal prediction efforts show very weak skill beyond one or two years and certainly don't concentrate on decadal (or 8-year) averages. So if the title should be kept, why not include a discussion of effects of different time averaging. Furthermore, in large parts of the analysis already different time-averaging is used and it is not mentioned how this relates to the main point of decadal predictability. It should also be mentioned that many of the hindcast systems used to evaluate decadal-scale forecast include observed solar irradiance. Moreover, forecasts of the strength of a solar cycle needed for actually deciding if a strong or weak solar forcing can be expected are far from being mature.

I will list a few more small issues in the following:

L11: "a systematic detection of solar-induced signals at the surface and the Sun's contribution to decadal climate predictability is still missing" Not clear what the authors want to say, here. Do they want to announce such a systematic detection in this paper? Certainly not, because they only do simulations. What would be a systematic detection? And is it at all possible with the available data?

L29: "forecast skill for several years (…) beyond the externally forced climate response (Smith et al., 2019)" I don't think this is an appropriate interpretation of the reference. Smith et al. are actually much more careful in the interpretation of their results.

L53: The sentence starting here is one of the examples where the remark that this is a result from a simulation is crucially missing.

S2: Information is missing on which simulations for which ensemble size are analyzed.

L94ff I guess correlation coefficients given here are only for a specific month. They seem to support a strong epoch-high correlation story, but numbers for wind in December, e.g. would look very different.

L107 "Synchronization" of what?

L151 "We here show …" Another case where the authors should mention that this refers to model reality.

L188 I think that good studies don't necessitate such "first time" claims but results speak for themselves. Moreover, with model simulations this problem in observations can't be overcome.

L195 "The solar cycle enforces the NAO phase." Even in these simulations, solar cycle forcing just changes the probability of occurrence of some phase.

---

## Author Comment (AC1)

*Reply to*
Review of "The Sun's Role for Decadal Climate Predictability in the North Atlantic"
submitted by Drews et al. for publication in ACP

**We thank the reviewer for the helpful comments.**
**Please find our replies and comments in italics.**

The authors interpret two ten-member ensembles of CMIP-type "historical" simulations with respect to solar cycle signals in the boreal winter and in particular the North Atlantic surface climate. I think that simulations and their analysis are very appropriate for the purpose, writing and figures are in general very clear, such that I expect the manuscript to become publishable after consideration of a few issues that I will elaborate below. Overall I qualify the, in my view, necessary revisions as major although they may require relatively little effort. My issues with the current form of the manuscript are largely related not to the results themselves, but to their interpretation, discussion and framing.

The authors write that they intend "to partly rebut the conclusions" of a study by Chiodo et al. (2019) which speaks in its title of an "insignificant influence of the 11-year solar cycle on the North Atlantic Oscillation". My understanding is that the authors don't question the observational part of the analysis of that paper, but the analysis of observations which was done for DJF means, while in this manuscript, a surface signal significant at the 90% level is detected for February. The authors conclude "that it might be necessary to analyze monthly fields to capture a highly monthly varying signal". Why speculate? I suggest to either analyze the Chiodo et al. model output for monthly signals and/or the new simulations for DJF signals.

***Additional analysis for solar signals in DJF:***
*As suggested by the reviewer, we did additional analysis for zonal mean temperature, zonal wind (Fig. R1) and SLP (Fig. R2) as DJF means during the strong epoch. As shown in Fig. R1a, the solar signals in DJF mean zonal temperature are consistent with the monthly results in our paper (Fig. S4), primary warming in the tropical stratopause and secondary warming in the lower stratosphere (above the tropical tropopause). Similar westerly winds also can be found in the DJF mean (Fig. R1b) in the polar vortex region. However, only when using monthly data the top-down propagation and a significant surface signal can clearly be seen (Fig. S5).*

[Figure]

**Fig. R1**. *Composite differences between the solar maximum and minimum for (a) zonal mean temperatures (K) and (b) zonal mean zonal winds (m/s) in DJF mean during the strong epoch. Significance levels are indicated by white dots (95%) based on 1000-fold bootstrapping test.*

*The North Atlantic surface signals for DJF mean during the strong epoch are shown in Fig. R2. There is a positive NAO pattern similar to that in February (Fig. 4a) but with smaller values. As shown in Fig. R1, the dipole zonal winds anomalies (easterly winds in the subtropics and westerly winds at high latitudes) in the troposphere are much stronger and significant in February, and smoothed out in the DJF mean. Accordingly, responses of the SLP and surface winds in the DJF mean are much weaker as compared to February.*

[Figure]

**a. SLP / Wind_850hPa in DJF (Strong)**

***Fig. R2.** Composite differences between the solar maximum and minimum for SLP (contours) and wind at 850hPa (vectors) in DJF mean. Only those vectors where the zonal wind component is significant at the 90% level are shown. White dots indicate 90% statistically significant level based on 1000-fold bootstrapping test for SLP.*

*We now added a sentence and the new figures referring to the new DJF mean analysis to the manuscript:*

*"Our results (see also Figs. S4-5) suggest that it might be necessary to analyze monthly fields to capture the top-down propagation of the solar-induced wind anomalies and surface signals. Analyzing our model using DJF means, we only find a very weak signal in SLP and no significant zonal mean zonal wind signal at the lower troposphere during the strong epoch either (Figs. S10-11)."*

The authors mention further differences, as the missing low-frequency solar variation in the Chiodo et al. simulations. But why should this matter given that the authors make an effort to exclude this part of the solar signals from their analysis. More generally, it would be appropriate to discuss the agreement or disagreement of this study's result with those of other papers more carefully. It seems the authors see their study in agreement of other papers they cite, e.g. in the Discussion "(Gray et al., 2010, 2013, 2016; Kodera, 2003; Kodera et al., 2016; Matthes et al., 2006)", which they oppose to Chiodo et al.. However, several of those studies actually discuss North Atlantic surface signals only for DJF, so it seems to me that the actual results of this paper (no DJF signal) are rather similar to those of Chiodo et al., and it is mostly the framing where it differs.

*We thank the reviewer for pointing out that it might be less the pure results, but rather the interpretation that has led to apparently different study results in the past. We now added a sentence to the end of the first paragraph in the Discussion section:*

*"However, as we show here, these seemingly discrepant results could be due to the analysis of DJF means in most studies, which likely are not sensitive enough to capture the signal reliably (see below)."*

My second major point is that I think the authors do not adequately compare their model results to the available observation-based datasets. In the Discussion they mention that they "do not find a lagged NAO response... while the largest response in the observations appear at a lag of two years." However, in the results section, when discussing Fig. 5c this is much less clearly presented. They write, e.g., that "running correlation ... begins to rise in the 1920's both for the model and

observations." But this is not at all the case for the observations analyzed for lag zero. The following sentence is probably unintentionally unclear about "lag of two years" relating to model or observations or both. Why not, to compare apples and apples, also include model results for lag +2 in Fig. 5c? Moreover, it would be useful to identify also for other analyses if the observations are in the range of results provided by the individual ensembles, even if, unfortunately, one can neither conclude with certainty from such an agreement that the model is correct, nor that the observations provide a typical signal.

*We agree that that sentence was a bit misleading. We now adjusted it to:*

*"Their running correlation for all overlapping 45-year windows is fluctuating in the earlier years but begins to rise in the 1920's both for the model (at 0 lag) and observations (with a lag of 2 years) (Fig. 5c)."*

*Furthermore, we would like to note that other studies did not find a lag in their model simulations either, and we therefore added a reference when addressing the lag differences in the Discussion:*

*"We do not find a lagged NAO response in our simulations (cf. Gray et al., 2013) [...]"*

*We attach here the figure of the running correlation of the model at lag 2, however, we would leave the figure in the manuscript as it is since it appears very full, and we clearly state that the model shows the strongest response at 0 lag.*

[Figure]

*Regarding the comparison of our analyses with observations, we add here the figure of the February NAO station-based index for all ensemble members and*

*observations, which shows that the observed index is well within the range of the model members.*

[Figure]

*Furthermore, we use ERA5 to show zonal mean temperature and zonal mean zonal wind for comparison, and provide these figures below as well as in the new version of the supplement. Please note that the available data for zonal wind and temperature 1) are not as high as we plotted in our study, 2) it only covers three solar cycles (1979-2015), and 3) the solar forcing in ERA5 is a constant value.*

*The solar maximum years for these composite are: 1980, 1981, 1982, 1989, 1990, 1991, 2000, 2001, 2002.*

*The solar minimum years are: 1985, 1986, 1987, 1995, 1996, 1997, 2013, 2014, 2015.*

[Figure]

**Fig. R3.** *Same as Fig. R1, but for observational DJF mean ERA5 (1979-2015).*

[Figure]

**Fig. R4.** *Same as Fig. R3, but for monthly ERA5 from Oct to Feb (1979-2015) (similar to Figures S4 and S5).*

Already in the abstract, the authors claim to "show that a strong solar cycle forcing organizes and synchronizes the decadal-scale component of the North Atlantic Oscillation". They use this formulation a few more times. I claim they only show that this is true in their model reality. Of course, this is very useful, and the same mechanisms could also act in reality, but we can't be sure. This in particular the case because of the involved non-linearities in the system, mentioned several times by the authors, and the apparently very different response to only slightly different forcings (compare weak and strong forcing epochs).

*We agree with the reviewer as far as we consider our study alone. However, we would like to make the point here that our study's result in this respect is not isolated but fits into a large collection of studies making use of different models and observational/reanalysis data. The understanding of the so-called top-down mechanism inducing a surface signal over the North Atlantic projecting onto the North Atlantic Oscillation is widely accepted in the research community dealing with this topic. We have to state that our study does not add much to the process understanding in this respect but we provide new insights as to why (when) this solar-related surface signal temporarily is hard to detect and what relevance this may have for (decadal) climate prediction efforts.*

*However, the reviewer is of course right, that we did not employ other models and no sophisticated analysis of observational/reanalysis data, hence we modified the respective statements in our manuscript to:*

*"The extratropical North Atlantic is a hotspot of solar cycle influence on climate predictability (Fig. 1a) where up to 25% of the decadal variability of winter surface air temperatures are explained by the solar cycle in our model."*

*"We here show that in our model this "organization" depends on the solar cycle amplitude and it is large enough for a potential predictability variance fraction (ppvf) of up to 25% in the North Atlantic region."*

*"We demonstrate that in our model the strength of the solar surface signal depends on the amplitude of the solar cycle."*

My last general point concerns the interpretation of the results with respect to decadal predictability. If this is supposed to be the main point, as the title suggests, I think this needs more careful and enhanced discussion. For instance, the authors claim that they use 8-year averaging because this is "a typical target of actual decadal prediction efforts". However, in the reference they mention for this (Goddard et al., 2013) it is said that their choice of 1, 4, and 8 years may seem arbitrary, but was chosen to illustrate the effects of different temporal averaging. Many actual decadal prediction efforts show very weak skill beyond one or two years and certainly don't concentrate on decadal (or 8-year) averages. So if the title should be kept, why not include a discussion of effects of different time averaging. Furthermore, in large parts of the analysis already different time-averaging is used and it is not mentioned how this relates to the main point of decadal predictability. It should also be mentioned that many of the hindcast systems used to evaluate decadal-scale forecast include observed solar irradiance. Moreover, forecasts of the strength of a solar cycle needed for actually deciding if a strong or weak solar forcing can be expected are far from being mature.

*Thank you for this comment. It tackles a number of issues and for reasons of clarity we will sub-divide our response here into bullet points:*

- *First, we would like to stress that the reviewer's statement about the lack of skill in decadal prediction systems beyond the first two forecast years is only true when individual years (or even seasons) are considered. When multi-annual averages are considered - something that is recommended by Goddard et al. (2013) and usual practice in decadal prediction verification - the skill (as measured e.g. by the correlation) for near-surface temperature is typically the higher the longer the temporal averaging interval. That means typically that the skill for a year 2-9 prediction is higher than for the respective year 2-5 prediction. Of course this is not a result of higher predictive accuracy in the later forecast years but just an effect of eliminating more "noise" by averaging over a longer time. And this relates to the fact that the source of this skill (for temperature) is predominantly found in external forcing. That means that the predictions are highly skillful, it is just that the benefit from initialization is comparably limited. However, there are a few regions where a*

number of studies suggest predictability beyond 5 or even 10 years despite the absence of strong externally driven signals. The most prominent example for this is the North Atlantic (see, e.g., Christensen et al., 2020).

- Second, the reviewer is correct that there are quite different temporal averaging intervals used in decadal prediction studies and in recent years, most studies rather focus on lead times up to five years. However, there are still studies being published that deal with longer lead times and averaging periods, a very prominent example being the probably most-cited decadal prediction paper in recent years, that is the study of Smith et al. 2019 (analyzing year 2-9 predictions for the boreal summer season). Further examples are, e.g., Meehl et al., 2014; Kadow et al., 2016; Borchert et al., 2021; Hu & Zhou, 2021; Tian et al., 2021.

- Third, the reviewer correctly mentions that we use different time averaging in our analysis. We have to admit that this is not really done on purpose but rather the effect of bringing together skill-oriented analysis meant to match practices in climate prediction studies (as in Fig. 1, using the 8y running mean) and process-oriented analysis (as in Fig. 3 & 5) which were inspired by other studies such as Thieblemont et al. (2015). Our response to Reviewer 2 includes a sensitivity study to prove that the ppvf results presented in Fig. 1 are not sensitive to the averaging window. We therefore suggest adding the following sentence to the Methods section, subsection "Decadal potential predictability": "Other window lengths between 7 and 10 years were tested to exclude sensitivity of the results depending on the averaging period; the results were very similar for all window lengths." We just show this in a band around the currently used 8 year smoothing. If we would use a considerably shorter filtering window like 3 years, it is clear that some results would change. We would expect especially a much higher ratio of internal variability in the tropical Pacific, as a 3y running mean would not filter out ENSO-variability anymore. However, we consider a detailed discussion of the effect of different averaging windows beyond the scope of this paper.

- Fourth, the reviewer is correct that the observed solar forcing is included in decadal prediction systems. However, the story is quite complex. From our point of view it is highly questionable how much of the predictability indicated in our study can be exploited by today's decadal prediction systems. In order to model a proper representation of the top-down mechanism being associated with the surface climate signals seen in our study (and many others) over the North Atlantic requires interactive chemistry modeling (or at least an ozone forcing incorporating effects of the 11y solar cycle), a model top well above the stratopause (approx. 1 hPa), a sophisticated short-wave radiation scheme in the model, and the usage of spectral solar irradiance (SSI) as forcing dataset to account for the higher variability in the UV part of

*the solar spectrum compared to the visible and near-infrared part. The CESM1(WACCM) model used by us for this study fulfills these requirements, plus incorporating a parametrization of energetic particle precipitation and respective forcing. Most of today's decadal prediction systems fulfill only parts of these requirements. However, in this respect we notice a large development from CMIP5 - containing the first coordinated decadal prediction exercise - to CMIP6. The recommended forcing datasets (solar and ozone) for CMIP6 provide everything required above. This is why we speculate that some prediction systems in use for CMIP6-DCPP (and their counterparts used for CMIP6-historical) may be able to model some representation of the top-down mechanism, particularly those with a model top beyond 1 hPa, a state-of-the-art short-wave radiation scheme, and actually using the transient SSI-forcing as well as the ozone forcing for CMIP6. The study of Borchert et al. (2021), which we cite in the discussion of our manuscript, points out that the multi-model ensemble of CMIP6 simulations exhibits a response to external forcings that better matches the observed temperature evolution particularly in the North Atlantic and that this can be attributed primarily to the response to natural external forcings such as volcanic aerosols and solar forcing. A further distinction was not possible with the experiments analysed by Borchert et al.*

*However, there are still a number of models used in CMIP6 and DCPP that are far from fulfilling the requirements to represent the top-down mechanism. Some still use the total solar irradiance (TSI) only as solar forcing, scaling temporal TSI variability homogeneously over all parts of the spectrum which leads to unrealistically high variability in the visible and near-infrared and too low variability in the UV part of the spectrum. The result is a dampening of any potential top-down mechanism in these models.*

*We may inform the reviewer about the fact that the authors participate in the ongoing research project "SOLCHECK - Solar contribution to climate change on decadal to centennial timescales" - funded by the German Federal Ministry for Education and Research - with one of the main aims to particularly quantify the solar contribution to prediction skill in a state-of-the-art decadal climate prediction system (based on MPI-ESM).*

- *Fifth, the reviewer is correct that the prediction of the 11y solar cycle itself is still a scientific challenge but positive prospects are given (see e.g. Petrovay, 2020). Furthermore, predictions of the progression of a cycle that already started are feasible within reasonable error margins and are actually produced by, e.g., the Solar Cycle Prediction Panel representing NOAA, NASA and the International Space Environmental Services (ISES), see [https://www.swpc.noaa.gov/products/solar-cycle-progression](https://www.swpc.noaa.gov/products/solar-cycle-progression). Therefore, we argue that the incorporation of predicted solar variability in quasi-operational decadal climate prediction may potentially be useful. Additionally, we consider the knowledge about a solar contribution to climate predictability valuable*

*despite limitations of solar cycle prediction skill. This knowledge improves our overall understanding of climate predictability and shows ways to potentially improve decadal climate prediction, depending on the actual skill of (future) solar cycle prediction.*

*Christensen, H. M., Berner, J., & Yeager, S. (2020): The Value of Initialization on Decadal Timescales: State-Dependent Predictability in the CESM Decadal Prediction Large Ensemble, Journal of Climate, 33(17), 7353-7370, DOI:* *https://doi.org/10.1175/JCLI-D-19-0571.1*

*Petrovay, K. (2020): Solar cycle prediction. Living Rev Sol Phys 17, 2, DOI:* *https://doi.org/10.1007/s41116-020-0022-z*

*Borchert et al.: Skillful decadal prediction of unforced southern European summer temperature variations. Environ. Res. Lett., 16, 104017 (2021)*

*Hu, S. & Zhou, T.: Skillful prediction of summer rainfall in the Tibetan Plateau on multiyear time scales. Science Advances, 7(24), eabf9395 (2021)*

*Kadow, C. et al.: Evaluation of forecasts by accuracy and spread in the MiKlip decadal climate prediction system. Meteorol. Z., 25, 631–643 (2016)*

*Meehl, G.A. et al: Decadal Climate Prediction: An Update from the Trenches, Bulletin of the American Meteorological Society, 95(2), 243-267 (2014)*

*Smith D.M. et al.: Robust skill of decadal climate predictions. npj Climate and Atmospheric Science, 2, 13 (2019)*

*Tian, T., et al.: Benefits of sea ice initialization for the interannual-to-decadal climate prediction skill in the Arctic in EC-Earth3, Geosci. Model Dev., 14, 4283–4305 (2021)*

I will list a few more small issues in the following:

L11: "a systematic detection of solar-induced signals at the surface and the Sun's contribution to decadal climate predictability is still missing" Not clear what the authors want to say, here. Do they want to announce such a systematic detection in this paper? Certainly not, because they only do simulations. What would be a systematic detection? And is it at all possible with the available data?

*We agree that this sentence starts a little unclear. We suggest changing it and writing the following: "Despite several studies on decadal-scale solar influence on climate, a systematic analysis of the Sun's contribution to decadal surface climate predictability is still missing." We would like to stress in that context (and refer additionally to our response to major comment 6 of Reviewer 2) that "predictability" is*

*just a theoretical concept and commonly estimated based on model simulations alone.*

L29: "forecast skill for several years (...) beyond the externally forced climate response (Smith et al., 2019)" I don't think this is an appropriate interpretation of the reference. Smith et al. are actually much more careful in the interpretation of their results.

*We are not entirely sure in how far the reviewer considers our reference here not being appropriate. Smith et al. (2019) claim that several other studies found a rather small benefit from initialization (i.e., beyond externally forced climate responses) mainly because their analysis approach and significance testing was not optimal. They further present a new approach assessing the impact of initialization in the residual space after subtracting the ensemble mean of uninitialized simulations from the initialized hindcasts as well as the observations. By doing so they find statistically significant benefits from initialization for a number of larger regions, e.g., over large parts of the Atlantic Ocean, Europe and parts of Africa, the Indian Ocean, Eastern Asia and parts of the Pacific, close to the Kuroshio (Extension). One issue that additionally needs to be considered though is that Smith et al. (2019) use a large multi-model ensemble of unprecedented size in this context. We suggest to rephrase the sentence in our manuscript to: "These prediction systems show forecast skill for several years (Bellucci et al., 2015; Yeager and Robson, 2017) and at least for large multi-model ensemble systems a robust benefit from initialization that goes beyond the externally forced climate response can be shown for number of regions globally (Smith et al., 2019)."*

L53: The sentence starting here is one of the examples where the remark that this is a result from a simulation is crucially missing.

*Agreed. We added "in our model" to this sentence.*

S2: Information is missing on which simulations for which ensemble size are analyzed.

*Agreed. We added the table with the models and ensemble sizes to the Supplement.*

L94ff I guess correlation coefficients given here are only for a specific month. They seem to support a strong epoch-high correlation story, but numbers for wind in December, e.g. would look very different.

*We now mention that it is December zonal mean zonal wind: "The ensemble mean zonal wind (here: December) gets more organized and in phase with the solar forcing during the strong epoch [...]"*

*To be specific, these are all correlation coefficients for DJF mean and December. "Var." is the variance fraction of the solar-induced changes compared to the magnitude of internal variability. "R" is the correlation coefficient with the solar forcing index.*

|  | *Weak epoch* | *Strong epoch* |  | *Weak epoch* | *Strong epoch* |
|---|---|---|---|---|---|
| **T_DJF** | Var. = 31% | Var. = 69% | **U_DJF** | Var. = 22% | Var. = 23% |
|  | R = 0.55 | R = 0.72 |  | R = -0.12 | R = 0.36 |
| **T_Dec** | Var. = 31% | Var. = 37% | **U_Dec** | Var. = 18% | Var. = 24% |
|  | R = 0.21 | R = 0.58 |  | R = -0.14 | R = 0.33 |

L107 "Synchronization" of what?

*We now added "of the decadal NAO phase" to the previous sentence and hope this makes it clearer: "We find the "typical" downward propagation of zonal wind anomalies in later winter (Fig. S5) and a synchronization of the decadal NAO phase of the ensemble members (Fig. 5)."*

L151 "We here show ..." Another case where the authors should mention that this refers to model reality.

*Agreed and "in our model" added.*

L188 I think that good studies don't necessitate such "first time" claims but results speak for themselves. Moreover, with model simulations this problem in observations can't be overcome.

*We exchanged this by "With this unique dataset, ...".*

L195 "The solar cycle enforces the NAO phase." Even in these simulations, solar cycle forcing just changes the probability of occurrence of some phase.

*We modified this sentence and it now reads: "This means the solar cycle enhances the probability of a specific decadal NAO phase if the solar forcing is strong enough."*

---

## Author Comment (AC2)

***Reply to Reviewer 2***

***Review of "The Sun's Role for Decadal Climate Predictability in the North Atlantic" submitted by Drews et al. for publication in ACP***

*We thank the reviewer for the comments. Please find our replies below in italics.*

The paper reports on the impact of the solar cycle on decadal predictability of the NAO in the WACCM chemistry climate model. The paper claims that the solar cycle "organizes" and "synchronizes" the decadal-scale component of the NAO. Based on these results, the paper concludes that the solar cycle substantially contributes to "potential predictability" of up to 25% in the North Atlantic, but that the solar influence is limited to decades with sufficiently strong solar cycle amplitudes, such as the second half of the 20th century. The subject of the paper is of relevance and interest for the readership at Atmospheric Chemistry and Physics, and the paper is well written. However, the evidence provided in support of the main two claims (1. the contribution of the solar cycle to predictability, and 2. the impact of the solar cycle amplitude) is not convincing, due to the convoluted and not sufficiently justified methods. Just naming a few examples: 1. the use of ppvf technique to quantify the "predictability", 2. the use of 90% significance level instead of the more standard 95% level, 3. the smoothing of the data, 4. the low and regionally limited significance of the results, etc... Hence, the implications of this paper concerning the potential prediction skill from the solar cycle are over-stated. The authors need to substantially revise some of these claims, perhaps toning them down and/or provide more convincing evidence in support of a robust and detectable solar signal. I also find the discussion rather biased at times. As examples: 1. the signals over the Atlantic are rather regionally very limited and yet the paper makes a big deal out of the small solar signals, 2. (1000) one THOUSAND years of model data are needed to get a "robust" solar signal (this amount of data means the signal can hardly be useful for decadal prediction...). Hence, I cannot recommend publication in the present form. Extensive revisions are needed, as detailed below.

MAJOR POINTS

1. The authors "disentangle" the solar-induced climate response from internal variability, using convoluted and not fully justified statistical methods. First, they show the "potential predictability" (Fig.1a) by using the ppvf technique on the **difference** between FULL and LOWFREQ, divided by the variance in FULL. Further, they smooth the data using an 8-year window, citing Goddard et al., to justify such choice (a paper which by the way does in no way suggest one should use 8 years). This metric does not effectively show the impact of the solar cycle on the variance itself (which is what the authors are after) but rather the relative impact of the solar signal over that of the long-term trends in the solar forcing. A much easier (and more convincing) metric would be the ratio between the variance in FULL and

LOWFREQ. Also, using an 8-year smoothing window to get an 11-year signal seems a very unwise way to filter the data. Why exactly are 8 years used? I suspect the results are sensitive to the window length used to smooth the data, but maybe the authors can prove me wrong.

***Reply***:

*The reviewer mentions a number of points here. For reasons of clarity, we structure our response in bullet points:*

- *The reviewer states that the analysis of the difference between the FULL and LOWFREQ ensembles is a convoluted and not justified statistical method. We would like to stress here that this approach is a standard method used in detection and attribution efforts. We cite "The Detection and Attribution Model Intercomparison Project (DAMIP v1.0) contribution to CMIP6" by Gillet et al. (2016) (reference list at the bottom): "There are two possible frameworks for designing climate model experiments for D&A analysis: the "only" approach, in which simulations are driven with changes only in the forcing of interest, while all other forcings are held at pre-industrial values; and the "all-but" approach, in which simulations are driven with changes in all forcings except the forcing of interest." (p. 3687). Further it is stated that "[...] to detect the contribution of a particular forcing to observed climate change [...] the "all-but" approach may be best [...]." and shortly thereafter: "For instance, the response to anthropogenic forcing can be diagnosed from planned DAMIP experiments by taking the difference of the historical and historical natural-only experiments, an "all-but" design [...]" (p. 3689). When combining the FULL- and LOWFREQ-ensembles by analysing the difference of the two ensemble means, we directly follow this "all-but"-approach to examine the effect of the 11y solar cycle. Using the ensemble mean extracts the response of the external forcing, so the difference between the ensemble mean of FULL and LOWFREQ is a way to extract the response to the 11-year solar cycle (see the detailed description in the "Methods" section). As described in the Methods section, first, we disentangled the solar cycle component and calculated its variance, then divided it by the total variance to obtain the ppvf.*
*We think that it is a very strong statement by the reviewer that such an approach developed by an international scientific community, used in a large number of peer-reviewed studies, and being an essential element of a model intercomparison project endorsed for CMIP6 is a not fully justified method. Given this background and the fact that the reviewer provides no further reasoning why this method should not be fully justified in her/his opinion, we decide to follow the approach as outlined and described in our manuscript.*
- *When asking for the ratio between the variances in FULL and LOWFREQ, the reviewer assumes that the total variance of the parameters analysed in this study is enhanced by adding an additional external forcing (the 11y solar cycle variability). This is not the case. The total variances of Ts in FULL and LOWFREQ are almost the same for both interannual (Fig.R1a-b) and decadal timescales*

*(filtered by 8-year running mean, Fig.R1c-d), as shown in Fig. R1. The total variance is defined as the variance of all the 10-members of each set (FULL and LOWFREQ, concatenate the 10-members to a large set and then calculate the variance). As stated in our manuscript, the solar cycle forcing rather acts as some kind of (weak) pacemaker, phase-locking (or synchronizing as it is called in Thieblemont et al.) large-scale variability patterns that also exist in the climate system (and its model representation) without external (solar) forcing.*

- *The reviewer states that an 8y-running mean is an unwise filter when analysing the 11y solar cycle. We refer the reviewer to Fig. R2, showing the filter characteristics for running means of different window lengths. About 30% of a signal with periodicity of 11 years - such as approximately the case for the solar sunspot cycle - will pass this filter. For completeness we also add here Fig. R3 showing the solar index smoothed with different window lengths. We originally chose and often use the running mean of 8 years to identify solar cycle signals as this filter shows a signal transfer of around 0 for the spectral band from 3-5 years which is the preferred period of ENSO in the used model CESM1(WACCM) (see Marsh et al., 2013). A running mean of shorter window length, e.g., 5 years shows a higher signal transfer rate for a periodicity of 11 years but is also associated with a non-negligible (negative) transfer rate in the ENSO spectrum. Using the 5y running mean would pose a significant risk of misinterpreting ENSO-related signals, especially in the tropical Pacific, as being of solar origin. Of course this argumentation is not necessary anymore once the ensemble mean of FULL, LOWFREQ or the difference between the two is analyzed. In this case, internal ENSO variability in the individual runs should be extensively eliminated by the ensemble averaging. However, given that we see no significant drawbacks of the 8y running mean, we prefer to leave the analysis as is.*

- *The reviewer suspects our ppvf results to be sensitive to the filtering with an 8y running mean. In Fig. R4 of this response letter, we provide the results of the very same analysis of the ppvf after applying a 7y running mean, an 8y running mean (as done in our study), and a 9y running mean. It is obvious that the results agree extremely well with each other, especially over the North Atlantic and surrounding areas. Hence, this point provides no reason either to change anything in our analyses or rephrase our description and interpretation.*

- *Furthermore the reviewer states that the paper of Goddard et al. "does in no way suggest one should use 8 years". Let us elaborate. Goddard et al. write: "The verification of the temporal information in this framework is provided at different scales: year 1, years 2–5, years 6–9, and years 2–9. This set of temporal smoothing choices may seem somewhat arbitrary, but it represents a small set of cases that can illustrate the quality of the information for different lead times and temporal averaging. [...] The year 2–9 average represents decadal-scale climate [...]." (p.251). An average over forecast lead years 2-9 is an 8-year average. It is true that other averaging intervals are also mentioned and considered possible, however, the specific averaging intervals mentioned in this paper became something like a common practice in decadal climate prediction efforts. We refer the reviewer to a selection of climate prediction studies using this averaging interval (see e.g. Meehl*

et al., 2014; Kadow et al., 2016; Borchert et al., 2021; Hu & Zhou, 2021; Tian et al., 2021), including the probably most cited decadal prediction study in recent years by Smith et al. (2019).

[Figure]

**Fig.R1.** *Total variance of DJF mean surface temperature (Ts) from 10-ensemble members of (a) FULL and (b) LOWFREQ.; (c) and (d) are same as (a) and (b), but the Ts is filtered by 8-year running mean.*

[Figure]

**Fig. R2**: *Signal transfer of a sinusoidal signal with a given period for running-mean filters with different window lengths. The thick black dashed line marks a period of 11 years, approximately the period of the solar cycle analysed in our study. The two black dotted lines mark the spectral band from 3 to 5 years for which the power spectrum of ENSO in the CESM1(WACCM) model shows peak values.*

[Figure]

**Fig.R3.** *Time series of F10.7 (black) and its running mean with various windows (red: 7-year; blue: 8-year; green: 9-year; purple: 10-year). The 8-year filtered time series exhibits a correlation of 0.78 with the original time series, equivalent to an explained variance of approx. 61% (blue line). (For other running mean windows: (1) 7-year (red line): r=0.87, var=76%; (2) 9-year (green line): r=0.69, var=48%; 10-year (purple): r=0.56, var=31%).*

[Figure]

**Fig. R4:** *Potential predictability variance fraction (ppvf; explained variance) with respect to the DJF surface air temperature associated with the 11-year solar cycle after applying a 7y running mean (top), an 8y running mean (center; just as provided in Fig. 1a of our manuscript), and a 9y running mean*

2. Along the same lines, another line of evidence used in the paper to show a "solar signal" is the running mean correlation over time and against the solar cycle amplitude (Figs.5c-d). But instead of a canonical running mean correlation, we are seeing the correlation (against the solar cycle) of the **difference** between two (independent) ensembles! I find this an utterly confusing and strange metric. Why not simply looking at the running mean correlation itself, rather than the difference of two ensembles? I get it that there are other forcings at work too in the FULL ensemble and that the authors wish to extract the solar cycle component, but the solar signal should emerge from the (forced and unforced) noise... if it's of use for decadal prediction - in the real world, multiple forcings are at work and not only the solar cycle. Looking at the correlation of the ensemble mean itself would highlight how the NAO itself correlates with the solar cycle, rather than its 'solar derived' component (which is supposed to be represented by the FULL-LOWFREQ difference). Physically, it makes little sense to look at the correlation of the difference between two ensembles, if what we're after is quantifying how the solar cycle influences a specific variable.

*Reply: As noted above, the signal does not "emerge from noise" when considering the ensembles separately by looking at, e.g., their variances. Changes in the climate system in the period of interest are dominated by anthropogenic climate change and the solar signal cannot properly be extracted if not using methods that are able to extract and enhance the signal such as ensembles and differences between ensembles. It is more or less in parallel to the low-frequency increase of solar irradiance AND the increase of solar cycle amplitude that the anthropogenic influence on climate is becoming more and more important. That's why an approach like ours is necessary and as described above, it is a standard approach used in many detection and attribution studies.*

*Several previous studies that were able to find the solar signal in single model runs used the unrealistically high solar forcing (using data from the Spectral Irradiance Monitor (SIM) instrument on the Solar Radiation and Climate Experiment (SORCE) satellite, which later showed to be too high).*

*Furthermore, we point the reviewer to Figure S3, where we show the correlation of smoothed winter surface air temperatures from FULL and from LOWFREQ with observations (not the correlation of the differences), and the difference of these correlations. This figure gives a direct quantitative estimate of the benefit for decadal climate prediction, using a metric often used in this context. This benefit seen over the North Atlantic is definitely relevant and very positively received in the climate prediction community (personal communication during a number of workshops and conferences).*

*For the figure showing the running correlation of both ensembles separately, see below where Fig. 5c is discussed.*

3. The "emergence" of the solar signal in the strong solar cycle epoch is shown in Fig.5c, but the model vs observations comparison is intrisically flawed in this figure. First, a different lag is chosen for the model (lag 0) and the obs (lag 0 and 2). The model at 0 lag shows better agreement with the lag 2 in OBS data, so the lag 2 in the model should be compared with the lag 2 in the OBS. Moreover, the observational data (which is a "pure correlation") does not show the equivalent of the FULL-LOWFREQ difference but rather of equivalent of FULL alone, since it contains all the observed forcings... so this is not an apple vs apple comparison! This would be another argument in favor of using the actual running mean correlation in the FULL ensemble rather than of the FULL-LOW difference to show a detectable solar "impact". You could compare the running mean correlation of FULL and should be able to show that it's higher than in LOWFREQ to convincingly demonstrate that the solar signal "emerges" from internal variability (which is the claim of the paper).

**Reply:** *Please see below for our detailed reply (discussing Figure 5c).*

4. The paper claims that the strong solar cycles organize and even synchronize the decadal-scale component of the NAO. This is an over-statement, which is not sufficiently supported by the evidence provided in the paper. Rather, the paper shows a small influence of the solar cycle over very limited portions of the North Atlantic (Fig.1a) and not over Europe. ALso, the solar influence is 2x smaller than the "forced" component (Fig.1b) and of the internal variability (Fig.1c). Can we deem such this a "useful" source of skill, considering that 1000 years of data are needed? Also, the method to extract the solar signal (see major comment 2. above) seems a bit ad-hoc rather than a robust and critical assessment.

**Reply:** *The reviewer is right that 1) we show the influence of the solar cycle over parts of the North Atlantic, and 2) that the solar signal is small compared to responses to external forcings elsewhere. We would like to note that 1) our paper is about the solar influence in the North Atlantic, which is where the North Atlantic Oscillation is found, and 2) that even a small signal can be beneficial in decadal prediction efforts. In Figure S3 we show that including the solar cycle increases the correlation with observations (even over Europe even though not as pronounced as over the North Atlantic), which is a typical metric in decadal climate prediction. Also, in Figure 1, surface temperature is shown, whereas we also show the solar surface signal in terms of sea level pressure, and show that an SLP index (the station-based NAO-like index) is organized by the solar cycle. This SLP index has stations over Europe (see Figure 4a), and the associated pattern is not localized to the North Atlantic. Furthermore, regarding the amount of data needed, we would like to note that current climate prediction efforts produce retrospective forecasts (also called hindcasts) consisting of at the very least 10 ensemble members times 5 years of predictions times 55 initializations (a total of 2750 model years) to assess prediction skill (see Boer et al., 2016 for the CMIP6 DCPP protocol). Furthermore, it is common knowledge these days in the climate prediction community that we face the so-called signal-to-noise paradox (see Dunstone et al., 2016, 2018, Scaife and Smith, 2018) when analyzing climate prediction skill. The signal-to-noise paradox describes the fact that "a*

*climate model can predict the real world better than itself despite being an imperfect representation of the real world and a perfect representation of itself. Although this highlights a clear deficiency in climate models it also provides an opportunity to create skilful forecasts even with imperfect models by taking the mean of a very large ensemble in order to extract the predictable signal. Where the model signal-to-noise ratio is too small, the number of ensemble members needed to remove the noise and extract the signal will be larger than it would be if the signal-to-noise ratio were correct, and the amplitude of the resulting model predictable signal will be too small. Nevertheless, the ensemble mean signal may be highly skilful if it correlates with the observed variability [...]." (Smith et al., 2019). In other words: Just because we need thousands of model years to detect the signal does not necessarily mean that the respective signal in the real world is so small that we would need thousands of real years to detect it.*

5. The paper essentially rebuts another paper on this subject (Chiodo et al., 2019), but fails to discuss the reasons for the inconsistency. Chiodo et al. used the **same** climate model as this paper (WACCM) and a solar forcing which would qualify as "**strong solar cycle**" forcing **throughout 500 years** (i.e. repetition of the "strong" cycles 19-23). And yet, they found a signal which is time-dependent, much in the same way as Fig.5c of this paper. Hence, the solar cycle amplitude argument does not seem to hold in this study. I urge the authors to more explicitly state this inconsistency throughout the paper (in the abstract AND conclusions) and discuss possible reasons for disagreement, rather than "brushing off" any evidence against their claim, as they do e.g. in L180.

*Reply: The reviewer is right that Chiodo et al. used a similar version of the same climate model. However, we discuss slight differences of the model versions in the manuscript, which includes differences in the solar forcing itself. Here we particularly emphasize that the solar forcing for CMIP6 features stronger variability in the UV part of the spectrum compared to the NRLSSI1-forcing used by Chiodo et al.. In this context it is also worth noting that Chiodo et al. do very well find indications of the co-called top-down mechanism, they just state that the signal is too small compared to the large internal variability to be visible in averages over 500 years. When applying an idealized solar forcing featuring unrealistically strong UV variability, they do find a constant and statistically significant strengthening of the NH polar vortex in boreal winter that progresses down to the surface (see Supplement to Chiodo et al., 2019, particularly Sec. B & C as well as Supplementary Figure 11). Besides this stronger UV variability of the solar forcing used for our study, we also include the effects of auroral electrons as parametrized in CESM1(WACCM) via an index of geomagnetic activity. The influence of these particles - called the EPP indirect effect (Randall et al., 2006) - can result in significantly decreased ozone concentrations over the winter pole which may affect the strength of the polar vortex in late winter/early spring similarly as the radiatively induced top-down mechanism. These energetic particles are not at all considered in the experiments analyzed by Chiodo et al.*

*Furthermore, we suggested differences in the analysis might be another reason for coming to different conclusions. To be more specific: We see that it is necessary to analyze monthly fields instead of DJF means. In the new version of the manuscript we will add figures of DJF means of zonal mean temperature, zonal mean zonal wind and sea level pressure (Supplement new Figures S11 and S12), which still show signs of the solar signal, but much weaker and not significant.*

6. The paper argues about enhanced "predictability", but does not provide actual metrics of enhanced skill scores, when the solar cycle is included in the model predictions. Could the authors show that the model's ACC score values increase during the strong solar cycle epochs? If not, I am afraid that any claims about predictability are not supported and remain pure conjectures.

**Reply:** *The terms predictability and prediction skill are often used synonymously, also in the scientific literature. However, they basically refer to different things. It is nicely put into a single sentence by Boer et al. (2013), writing: "Climate predictability is a feature of the physical system that characterizes its "ability to be predicted" rather than "the ability to predict it" which is characterized by forecast skill." For a number of years it was common practice to assess predictability, or as it was more specifically called "potential predictability" by employing the "perfect model approach". That is the analysis of the ability of a model to predict itself, e.g., calculating a skill score for an ensemble mean predicting one of its individual members. This was historically understood as something like an upper limit of actual prediction skill with the very same prediction system. However, in recent years the climate prediction community learned about the above-mentioned signal-to-noise paradox. This essentially means that this assumption about potential predictability being the upper limit for prediction skill does not hold. Still, model-based potential predictability studies are done to derive some quantitative indication of where the climate system might be more or less predictable and in the context of trying to understand the underlying mechanisms. Anyway, actual prediction skill can be measured of course. And that is what we also provide in our manuscript. We refer the reviewer to Figure S3, which shows ACCs for FULL and LOWFREQ and the difference between them. This is a standard measure of deterministic verification used in decadal climate prediction. This fact is also discussed in our manuscript, saying: "This is further supported by Figure S3 when comparing the "skill" (correlation with observations) of FULL and LOWFREQ for the North Atlantic region. Consequently, solar variability and an adequate representation of its impact on climate is key to exploit the solar-induced potential predictability for decadal climate predictions." (lines 70-73 of the original submission). A comparison to the weak epoch analyzing a potential increase of the skill score as requested by the reviewer is not feasible as there are no observations of comparable reliability for this earlier period (before 1940).*

SPECIFIC ISSUES

L37 this is an over-statement. Given the problems outlined above, I find it hard to believe this is "robust evidence of solar influence". The solar influence is still minimal, so at very least, change this to "albeit small compared to internal variability" or something similar.

*Our interpretation regarding the robustness and value of these findings differs from the reviewer's opinion, however, we added the following sub-clause: "which is small but non-negligible compared to internal variability".*

L40 "relative to other external forcings" --> incorrect statement - other forcings have not been quantified, as you only compare solar against anything else.

*We mean and write: "relative to other external forcings and internal variability", which we consider to be "anything else" as the reviewer states.*

L41 realistic solar forcing --> whether it's realistic is quite debatable, as solar forcing still a reconstruction using statistical models, so please remove the "realistic" word

*First, using a statistical model does not contradict producing a realistic result (here the solar forcing). State-of-the-art models for solar irradiance, such as the NRLSSI2-model (Coddington et al., 2016) and SATIRE-S (Yeo et al., 2014), which were used to produce the solar irradiance forcing recommended for CMIP6, are able to reconstruct solar irradiance (spectral and total) with very high agreement to independent observations. However, we now write "most sophisticated solar forcing dataset".*

*As we have additionally been contacted by Martin Andrews, who reminded us of his paper, we extended and rephrased the whole sentence to "A similar experimental approach has been used by Andrews et al. (2015), however, our simulations are much longer and include the most sophisticated solar forcing dataset currently available for climate models and recommended for CMIP6 (including solar radiative and particle forcing), a well-resolved shortwave radiation scheme, and a comprehensive module for middle atmosphere chemistry modelling."*

L50 extracting an 11-yr signal using an 8-yr smoothing window is risky, as the window is close to the frequency you are interested in. Further, the paper by Goddard et al does not justify the use of 8-yr smoothing for the study of decadal signals. Are the results sensitive to this "smoothing"?

*Please see above (Figs. R2, R3 and associated text) for an elaboration of the effect of the window size. We had originally chosen this window size to eliminate noise and other modes of variability, especially ENSO, hence the window size could not be much smaller when analyzing individual simulations. Additionally, the 8-year smoothing is common in decadal predictions (average of years 2-9).*

L54-55 where is the 25% number coming from? Fig.1 shows signals over quite remote sections of the N.Atlantic rather than Europe itself, and they are a small fraction of internal & forced variances. In any case, results do not seem to support this statement, as the authors haven't consistently shown that solar explains 25% of the decadal variance, but rather of a convoluted metric for the solar signal itself (i.e. the difference in FULL-LOW rather than FULL itself). I would urge authors to clarify how this number is obtained, or tone down this statement.

*To explain where the 25% are coming from, we provide Fig. R5. Here we show the ppvf associated with the 11y solar cycle, exactly as done in Fig. 1a of our manuscript, just zooming in over the North Atlantic and using a different color scale to make it easier to understand where our numbers are coming from. It is clearly visible that the ppvf peaks along the Southeastern coast of Greenland with maximum values in the (color) range of 0.24 to 0.27. That's why we write "up to 25%". For parts of the subpolar gyre, the ppvf exceeds 0.18.*
*The description of these signals as "quite remote sections of the N. Atlantic" given by the reviewer is not comprehensible from our point of view. It is clearly visible in this figure as it is in Fig. 1a of our manuscript that the ppvf associated with the solar cycle is statistically significant over large parts of the North Atlantic and Nordic Seas.*

*We write nothing in the text here about Europe, so we do not understand why the reviewer critically raises this point. Besides, it is also visible in Fig. R5 as well as in Fig. 1a that the ppvf associated with the 11y solar cycle is also significant over several parts of Europe.*

*In this specific sentence, we will adjust "decadal variability" to "decadal variance".*

*We will not discuss the reviewer's claim of our method being convoluted again as we already made clear that this method is standard and widely accepted in the context of detection and attribution studies.*

[Figure]

**Fig. R5**: *Potential predictability variance fraction (ppvf; explained variance) with respect to the DJF 8-year averaged surface air temperature associated with the 11-year solar cycle; this is exactly what is plotted in Fig. 1a of our manuscript, just zoomed in over the North Atlantic and using a different color scale*

55-56 isn't this sentence in complete contradiction with what is stated in previous sentence? If this region is low in terms of predictability, then how can one say solar cycle influence on climate predictasbility ? How is the significance at all quantified?

*We state here that predictability associated with the solar cycle is comparably high in this region, but stemming from other external forcings and from internal variability, the same region shows low predictability.*

*Regarding the significance testing, we refer the reviewer to the Methods section. Our manuscript is written in a letter format, hence, it has a rather short word limit in the main body of the text with the requirement to provide the Methods section at the end of the manuscript.*

Fig.1: how valid is it to disentangle the solar signal using the difference FULL - LOWFREQ? The authors are using a linear estimation for something which is intrinsically nonlinear, as they state. Can we also assume variances in FULL and LOWFREQ are really the same to allow this quantification, or do they change?

To show the impact of the solar cycle on the decadal variability, the authors should rather show the ratios of the variance in LOW vs FULL - this would be easier to interpret and also more convincing evidence for a solar impact rather than the convoluted metric used here var(FULL-LOW)/var(FULL). Further, estimating the variance of the difference FULL - LOW over time does not make much sense physically, since it's inconsistent with the physical state of either of the two ensembles.

*Yes, the total variances in FULL and LOWFREQ are almost identical, see the reply to major point 1 above. They are dominated by the global warming signal. There is no linear additivity in the sense that the solar cycle increases the total variance of FULL as compared to LOWFREQ, hence, showing the ratio of variances of LOWFREQ and FULL would not yield anything else but noise.*
*The applicability of the general approach to analyze the difference between FULL and LOWFREQ in the sense of a typical detection and attribution study is already discussed in our answer to major comment 1.*

Fig.1a shows that actually, the decadal predictability is quite limited regionally and does not extend to the European continent. But more generally, how can we get statistical significance on something which is 2x smaller than internal variance? How is the significance level effectively estimated in the ppvf technique?

*Statistically insignificant regions (p>0.05) in Fig. 1a and b are estimated by means of a Fisher's f-test, see the "Methods" section.*
*We don't understand why the reviewer considers a ppvf signal over Europe being a criterion for relevance or not. However, significant ppvf values are also evident over several parts of Europe, namely the Iberian Peninsula and France as well as Western Scandinavia. This is very well visible in Fig. 1a of our manuscript and also in Fig. R5 attached to this response letter. Relating this result to a potential direct effect on prediction skill, we once again refer the reviewer to Figure S3 which shows the difference in ACC (a standard deterministic skill metric used in climate prediction) with consistently positive values all over Europe.*

General remark: The ppvf is applied on separate runs, but the technique is not really well known in the climate community... so I would please ask the authors to explain better how they use it. Otherwise, the results will not be reproducible.

*We think that our description in the Methods section (Subsection "Decadal potential predictability") explains in detail how we calculated the ppvf and provide the reference to the study originally introducing this method (Boer, 2004). Following this initial study, the very same approach has been used by several other studies (see e.g. Boer & Lambert, 2008; Boer, 2011; Xu et al., 2020). Therefore, we consider the foundations of this method and our calculation reproducible following the description given in our manuscript.*

Fig.2 - this is a nice schematic, but there is literally nothing new here over e.g. the schematic by Gray et al., 2010 and the ones by Kodera et al. - hence, I frankly do not see the value of this figure and would recommend removing it.

*The reviewer is absolutely correct here and we do not intend to state that we invented something new here. We introduced this schematic to improve readability and reduce text - please be reminded that this manuscript is in a letter format and should be kept short. The schematic allows us to touch on the top-down mechanism only briefly, while reminding readers about it. We would therefore like to keep it. We will add "Inspired by Gray et al., 2010" to the figure caption.*

Fig.3b If the averaging over 10 ensemble members brings out the forced signal, then why is there not corresponding polar vortex strengthening around 1960, which is one of the strongest solar cycles on records? Further, why does the vortex at times even anticipate the solar cycle, such as e.g. at year 1980? Using multiple runs should bring out the signal even in individual cycles... so this should still work at all times!

*We cannot explain the behavior / response of the model during every single cycle as the polar vortex is highly dynamical and the solar forcing is only one component. To make this clearer, we suggest adding the following sentence to the manuscript behind previous line 98: "These numbers demonstrate that the polar vortex is highly dynamical and the solar forcing is only one component influencing it."*

*However, we would like to refer the reviewer to Figure S8. Here we show that the correlation between then F10.7cm solar radio flux index and the ensemble mean for highly dynamical parameters such as the polar vortex strength (indicated by the zonal mean zonal wind at 1hPa, averaged over the latitudinal band 55°N-65°N; Fig. S8b) or the NAO-index (Fig. S8c) is obviously not saturated for an ensemble size of 10 members. Such behaviour is nothing new. Murphy (1990) published a study on this behaviour including an approach to assess the correlation for a theoretical ensemble of infinite size. We also refer the reviewer to the study of Hansen et al. (2017) and particularly their Fig. 3 where they nicely illustrate the dependency of the correlation on ensemble size based on a seasonal prediction ensemble, analyzing NAO predictability. The Decadal Climate Prediction Project (DCPP) of CMIP6 defined 10 ensemble members to be the absolute minimum for participation (Boer et al., 2016), hence this ensemble size became standard for efforts in that context, still it is quite unique for a study based on a complex chemistry-climate model. Given the information about correlation growing with ensemble size, a solar influence with respect to a strengthening of a polar vortex might be detectable for even shorter periods or individual cycles if more members were available.*

Fig.5a What about the phase shifts in individual runs? Why does the sinus curve look shifted in some (2-3) of the runs? This rather hints at a sporadic & random process, rather than a "synchronization"... aLso, it does not make much sense physically to compare individual members in FULL against the ensemble mean of LOW, which is a separate ensemble!

*We cannot explain phase shifts of individual runs. For the 2-3 runs mentioned by the reviewer, the correlation at lag -1 year is only slightly higher than for lag 0. Please note that this figure still contains internal variability which is huge for this NAO-like index. It is the overall picture that counts: The members are much better aligned when the 11-year solar forcing is active. And the overall picture might look even better with more members (see reply to comment above).*

*Please note that we are not comparing the members of FULL against the ensemble mean of LOWFREQ, rather, we subtract the ensemble mean of LOWFREQ from the members of FULL in order to remove the signal of external forcing (except 11-year solar cycle) - this is the way we isolate the solar signal (see "Methods" section).*

Fig.5b What about the same calculation, for the individual LOW minus LOW-ensemble mean differences? This panel would be important to evaluate how much the unforced decadal variability itself can originate the apparent "synchronization"!

*As suggested by the reviewer, we plotted the crosscorrelation for the individual members of LOWFREQ, LOWFREQ ensemble mean removed, with the solar cycle, see below. As expected, we do not find any hint of synchronization without the solar forcing.*

[Figure]

Fig.5c what does "FULL-LOWFREQ" mean? Is this the running mean correlation between the solar index and the difference in the NAO at each year of the simulation, or is this the difference in the running mean correlation in FULL vs LOWFREQ? If the latter is the case, then it's not really a running correlation. I find this method quite convoluted and not fully justified. Why not simply looking at the running mean correlation itself, rather than the difference? This would highlight how the NAO itself correlates with the solar cycle, rather than its 'solar derived' component (which is supposed to be represented by the FULL-LOWFREQ difference)

*It is the running correlation of the solar index and the differences of the ensemble mean NAO indices as explained in the Methods section. We propose to add this info to the figure caption. We argue that we extract the solar signal by subtracting the LOWFREQ ensemble mean for any variable, i.e., here the LOWFREQ ensemble mean NAO index from*

*the FULL NAO index so that we get the solar signal in the decadal-scale NAO. Then we correlate this solar-induced NAO with the solar index.*

*For completeness, we add here the figure of the running correlations of the two ensembles (without subtraction of the LOWFREQ ensemble mean) and for both lags. Indeed we see that the only positive correlation arises in the FULL ensemble at lag 0, however, the LOWFREQ ensemble shows a negative running correlation of similar magnitude. We argue that this is because the solar signal is hard to detect when looking at the full data with all forcings included, hence, one needs to eliminate these other influences by subtracting the LOWFREQ ensemble mean.*

*That the solar signal is apparently harder to find in the model as compared to observations reminds of the above-mentioned signal-to-noise paradox. This does not mean that the signal is absent in the models - here, we would like to refer to the recent study by Smith et al. (2020) who were able to extract an NAO signal from what looks like model "noise".*

[Figure]

Fig.5c - why is only lag 0 shown for the model and not the lag 2, as done with the observations? The model shows better agreement with the lag 2 in OBS data... so the lag 2 in the model should be compared with the lag 2 in the OBS. Moreover, the observational data does not show the equivalent of the FULL-LOWFREQ difference but rather of FULL alone, so this is not an apple vs apple comparison! This would be another argument in favor of using the actual running mean correlation in the FULL ensemble rather than of the FULL-LOW difference to show a detectable solar "impact". You could compare the running mean correlation of FULL and should be able to show that it's higher than in LOWFREQ to convincingly demonstrate that the

solar signal "emerges" from internal variability (which is the claim of the paper) - see major comment above.

*We agree that our text was a bit unclear regarding the differences in lags in model and observations. We would like to adjust the text the following way: "Their running correlation for all overlapping 45-year windows is fluctuating in the earlier years but begins to rise in the 1920's both for the model (at 0 lag) and observations (with a lag of 2 years) (Fig. 5c)."*

*We would like to note that the 2-year lag has frequently been pointed out in many studies cited here, while the "missing lag" in models is not a new finding either (cf. Gray et al., 2013), and has already been attributed to insufficiencies of the models in ocean-atmosphere coupling (Scaife et al., 2013) - a point that we mention ("[...] (ii) that the model feedback from the ocean is insufficient [...]")*

*See also reply to comment above including the figure in question.*

Fig.5c - why is the dip in the model correlation around year 1900 not captured by the observations? Can the authors speculate?

*We interpret this in a way that during the weak epoch, there is no stable relationship between the solar cycle and the decadal-scale NAO, i.e., the dip occurs just by chance (after all, it is just a dip and not a feature that stays), and there is not necessarily something physically meaningful behind the dip. Furthermore, data quality during the weak period is rather poor.*

Fig.5d - if the running mean correlation is calculated over 45-y windows, then the individual data points are not mutually independent, and this would reduce the degrees of freedom. Is this taken into account in the calculation of the 90% error bar?

*Yes, the significance is tested taking serial correlation into account, see Methods section.*

Fig.5d If the scatter plot is for February, then the main conclusion about the "enhanced decadal-scale component of the NAO under a strong solar cycle forcing" only applies to this month, and not to, as previously argued in the literature, the whole boreal winter. This should be clarified in the abstract.

*In the supplementary Fig. S5 we show that the top-down mechanism with a strengthening of the polar vortex is very well visible in our model throughout November to January, too. The arrival of this signal at the surface is then to be seen in the February mean, according to our model experiments. This is very well in agreement with the common understanding of the top-down mechanism. However, we do not claim that the timing of this arrival is fixed. It is rather the result of interaction with internal variability and the seasonal cycle*

*and it could be slightly different in other models, the month that the signal arrives at the surface, and even the year that the signal is strongest (see above for our replies about the lag). We here trace the propagation using monthly fields and demonstrate that this is necessary to detect and follow the signal. We do, however, not want to state that this is the same for all models or the real world. We could add the following to the abstract: "[...] and that it might be necessary to look at monthly fields instead of winter averages to detect the solar signal at the surface."*

L148 "18% of the magnitude of internal variability" is again, a misleading statement, as the analysis using ppvf does not really quantify the magnitude of the internal variability on a specific time-scale.

*We agree that this sentence is quite misleading as parts of the information got lost during the editing process. What we mean here is that the variance of the solar-induced NAO index is only 19% (we apologize, there was also a glitch here about the correct number) of the magnitude of the variance of the "internal" NAO index (variance of the NAO indices of all 10 members minus the ensemble mean, which is supposed to be internal variability). This can nicely be seen in the following figure which shows all members and the ensemble mean.*

[Figure]

*We propose to extend the sentence in the following way:*

*"A comparison of the solar signal in the NAO with internal NAO variability, by calculating the variances of the smoothed ensemble mean index and those of the smoothed individual members, reveals that the solar signal in SLP over Europe is approximately 19% the magnitude of internal variability during the strong epoch."*

L150 "small in magnitude but manifests itself as an organization and synchronization of internal variability as shown by the cross-correlations" --> the cross correlation is not really a cross correlation in the cleanest statistical sense, but rather an ad-hoc construct designed to isolate the "solar signal" (FULL-LOWFREQ) rather than the solar influence itself. This would be e.g. more convincingly shown by providing evidence that the decadal variance in FULL and LOWFREQ are significantly different.

*As mentioned above, the total variances and decadal variances do not differ in the two ensembles, as they are dominated by the anthropogenic climate change signal. The solar signal is not detectable as an additive variance. We do, however, find it by looking at the phases of the decadal-scale NAO, which is what we show in Figures 5a and b.*

L180 - actually, it's not inconsistent, as the signal is not really significant at the 95% level here either (only at 90% level), which indicates that there's a (non negligible) probability that the signal may be by chance. I think this should be stated here.

*We do not write "inconsistent", we even write that our figure is quite similar to that shown in Chiodo et al. However, we interpret it differently.*

*Also, the choice of the significance level, be it 95% or 90%, is always rather arbitrary. With the 95% level, there is a non-negligible (i.e., 5%) probability for signals arising by chance, too. We state that we used the 90% significance level at several locations in the manuscript.*

L195 "enforces the NAO phase if the solar forcing is strong enough" - this is really hard to believe, as the phase is really not constant over time. Moreover, Chiodo 2019 also used a strong epoch for the solar forcing, and got a time-dependent signal, too. Hence, the authors should at the very least comment on that, and elaborate possible reasons for the disagreement.

*We could rephrase to: "This means the solar cycle enhances the probability of a specific decadal NAO phase if the solar forcing is strong enough. During solar maximum, there is a tendency for a positive decadal-scale NAO and vice versa."*

*In the paragraph below (previous ll. 197), we elaborate possible reasons for the disagreement with Chiodo et al., including the different solar forcing dataset and the analysis of monthly versus winter mean fields. In a new version of the manuscript, we added figures of DJF mean fields of our model results, which only show a very weak and non-significant signal at the surface.*

L215 interestingly, the disagreement between model and observations in terms of the lag is only noted here. Could it also be that part of the signal in the observations is by chance? Could this possibility at least be listed here?

*We agree that this cannot be ruled out and we suggest to modify the sentence the following way:*

*[...] (i) that the observational record is only one ensemble member that includes all internal variability and responses to all external forcings (which may even cause what appears as a lagged response, the lag itself being by chance) [...]*

L220 the pptf technique does not really convincingly demonstrate that the potential predictability is enhanced by this much (20-25%), as the signals over wide parts of

Europe remain insignificant (Fig.1) and there is no convincing demonstration that the skill of the model is improved over the decades with a "strong solar cycle". Hence, this remains an unjustified claim rather than a science-based statement. Rather, this analysis shows that a small solar cycle signal may be present, but that an enormous amount of model data is needed to make it statistically detectable.

*We explained in more detail already where the quantitative estimate of 20-25% over the North Atlantic are to be found. We would further like to refer the reviewer to Figure S3, which does show that skill (measured by the correlation here) increases over Europe. Therefore we consider our statement very well justified. Furthermore, it is widely known and accepted in the research community on decadal climate predictions that transient simulations subject to CMIP6-historical forcing do show a high degree of predictive skill for surface air temperature stemming from the external forcing (as shown for our ensemble in Fig. S3a) over many regions of the globe. It is also a fact that this level of skill is hard to beat by actual prediction systems, the comparison to CMIP6-historical simulations is always one of the hardest tests for climate prediction systems. The fact that we can show for our model ensemble that a substantial part of this skill can be attributed to the solar cycle is anything else but irrelevant for the climate prediction research community. Furthermore, these figures are only for surface air temperature, but from the NAO analysis it is clear that SLP changes over Europe when solar forcing is active.*

Also, if we need to run a model for 1000s of years to get a solar signal, then this would rather argue against an effective usability of the solar forcing for decadal prediction. Since this study does effectively not quantify the predictability (e.g. by using prediction skill scores metrics, or similar), I urge the authors to tone down any "predictability" statements.

*We consider the statement of the reviewer regarding the relevance of our findings and an "effective usability" to be an opinion. As already mentioned above, it is standard procedure - even part of the CMIP6-DCPP protocol - to run several thousand model years to assess climate predictability and prediction skill on decadal timescales. The need for this is partly rooted in the above-mentioned so-called signal-to-noise paradox which essentially means that the real world seems to be more predictable than model simulations suggest. We also emphasize once again that we do use actual skill metrics used by the climate prediction community, the Pearson correlation that is shown in supplementary Fig. S3. We explained already that the ppvf approach as well as the analysis of the difference between two ensembles are well established and documented peer-reviewed methods in studies on climate predictability and detection and attribution. We therefore see no need to tone down our statements about predictability.*

**References:**

*Boer, G. J., and Lambert, S. J.: Multi-model decadal potential predictability of precipitation and temperature, Geophys. Res. Lett., 35, L05706 (2008)*

Boer, G.J. Decadal potential predictability of twenty-first century climate. Clim Dyn 36, 1119–1133 (2011)

Boer, G. et al.: Decadal predictability and forecast skill. Climate Dyn., 41, 1817–1833 (2013)

Borchert et al.: Skillful decadal prediction of unforced southern European summer temperature variations. Environ. Res. Lett., 16, 104017 (2021)

Coddington, O. et al.: A Solar Irradiance Climate Data Record. Bull. Amer. Meteor. Soc., Bulletin of the American Meteorological Society, American Meteorological Society, 97, 1265–1282 (2016)

Dunstone, N. J. et al. Skilful predictions of the winter North Atlantic Oscillation one year ahead. Nature Geosci. 9, 809–815 (2016).

Dunstone, N. J. et al. Skilful seasonal predictions of summer European rainfall. Geophys. Res. Lett. 45, 3246–3254 (2018).

Goddard, L. et al.: A verification framework for interannual-to-decadal predictions experiments. Climate Dyn., 40, 245–272 (2013).

Hansen, F. et al.: Remote control of North Atlantic Oscillation predictability via the stratosphere. Q.J.R. Meteorol. Soc., 143: 706-719 (2017)

Hu, S. & Zhou, T.: Skillful prediction of summer rainfall in the Tibetan Plateau on multiyear time scales. Science Advances, 7(24), eabf9395 (2021)

Kadow, C. et al.: Evaluation of forecasts by accuracy and spread in the MiKlip decadal climate prediction system. Meteorol. Z., 25, 631–643 (2016)

Marsh, D.R. et al.: Climate Change from 1850 to 2005 Simulated in CESM1(WACCM). J. Climate, 26, 7372–7391 (2013).

Meehl, G.A. et al: Decadal Climate Prediction: An Update from the Trenches, Bulletin of the American Meteorological Society, 95(2), 243-267 (2014)

Murphy J.M: Assessment of the practical utility of extended range ensemble forecasts. Q. J. R. Meteorol. Soc. 116: 89–125 (1990)

Randall C.E. et al.: Enhanced $NO_x$ in 2006 linked to strong upper stratospheric Arctic vortex. Geophys Res Lett, 33, 18811 (2006)

Scaife, A. A. & Smith, D. M. A signal-to-noise paradox in climate science. npj Climate and Atmospheric Science 1, 28 (2018).

Smith, D.M., Scaife, A.A., Eade, R. et al. North Atlantic climate far more predictable than models imply. Nature 583, 796–800 (2020). https://doi.org/10.1038/s41586-020-2525-0.

*Smith D.M. et al.: Robust skill of decadal climate predictions. npj Climate and Atmospheric Science, 2, 13 (2019)*

*Tian, T., et al.: Benefits of sea ice initialization for the interannual-to-decadal climate prediction skill in the Arctic in EC-Earth3, Geosci. Model Dev., 14, 4283–4305 (2021)*

*Xu, L. et al.: Potential precipitation predictability decreases under future warming. Geophysical Research Letters, 47, e2020GL090798 (2020)*

*Yeo, K. L. et al.: Reconstruction of total and spectral solar irradiance from 1974 to 2013 based on KPVT, SoHO/MDI, and SDO/HMI observations. Astron. Astrophys., 570, A85 (2014)*

---

## Referee Report (RR1)

Review of "The Sun's Role for Decadal Climate Predictability in the North Atlantic" revised by Drews et al. for publication in ACP

I'd like to thank the authors for considering my earlier comments and the effort they put into providing additional material. In my view, the paper has improved significantly. There are however still some issues which I'd like to see considered.

Page numbers refer to the version with highlighted changes.

I'm still a bit at odds with the partly inconsistent comparison to the study by Chiodo et al. (2019; C19 hereafter). In L193 the authors speak of similar correlations but different conclusions in comparison to C19. In L193 they say the C19 study "revealed" insignificant responses. Should this refer to the analysis of observations or simulations? If the latter the word "reveal" seems to be at conflict with the above statement. Furthermore, later the authors list many reasons for differences between the simulations of this study and Chiodo et al. As the aim is to "partly rebut the conclusions" of C19 it would be important to be very clear. Are the simulation results really different? Or is it just a different interpretation of similar results.

I'm also confused by the new statement "seemingly discrepant results could be due to the analysis of DJF means in most studies, which likely are not sensitive enough to capture the signal reliably". Does this refer to the papers cited in the bracket above "(Gray et al. …)"? Why do they show signals if they used the not sensitive enough DJF means? If this refers to other studies, please cite.

L61 The authors contrast forced signals in the extratropical North Atlantic by saying that "up to 25% of decadal variance" is explained by the solar cycle and "this region shows low potential predictability due to other forcings. While this is not wrong it sounds like the solar influence is large compared to other forcings, which is not true. Averaging over this region by eye, I'd suggest that other forcings are still more important even in this region. It would be good to rephrase these sentences in order to avoid misinterpretation.

L80 "Consequently, solar variability and an adequate representation of its impact on climate is key to exploit the solar-induced potential predictability for decadal climate predictions." This sentence sounds odd. Of course solar variability is key to exploit the predictability potential created by itself, and how could it be done in models if the representation of its impact was inadequate.

L117 "This shows that the response to the solar cycle is highly non-linear and not necessarily proportional to the forcing." Isn't "non-linear: and "not proportional" the same? Why then the very different characterizations ("highly" vs. "not necessarily"?). I would agree with the latter but not the first statement. To show that the response is non-linear one would need to show that the response of the strong epoch scaled to the weak-epoch forcing is statistically significant different from the weak-epoch response.

L158 "the correlations reach statistical significances of 90% in the model, and in observations with a lag of two years " Although the different lags are mentioned in the preceding sentence, this sentence on its own can easily be misunderstood. The different-lag issue should be picked up again. Besides, I don't think that one more line (as kindly produced

in the response to my earlier review would make the figure too busy. So please include it. Furthermore, The correlations reach barely outside the 90% range. And, leaving aside autocorrelation issues, wouldn't one expect 10% of the values to be outside this range accidentally?

L163 "organization and synchronization of internal variability" Is organization different to synchronization? Please explain. And I'd say the NAO index is synchronized, internal variability remains internal variability.

L193 The meaning of the "hence" is not clear to me.

L207 "The NAO in turn is organized and synchronized by the solar cycle. "
Not sure why "in turn". Besides there is again the issue of organization and synchronization.

---

## Author Response (AR2)

*We thank the reviewer for carefully reading our paper and our suggested changes. Please find our further changes below in italics.*

Review of "The Sun's Role for Decadal Climate Predictability in the North Atlantic" revised by Drews et al. for publication in ACP

I'd like to thank the authors for considering my earlier comments and the effort they put into providing additional material. In my view, the paper has improved significantly. There are however still some issues which I'd like to see considered.

Page numbers refer to the version with highlighted changes.

I'm still a bit at odds with the partly inconsistent comparison to the study by Chiodo et al. (2019; C19 hereafter). In L193 the authors speak of similar correlations but different conclusions in comparison to C19. In L193 they say the C19 study "revealed" insignificant responses. Should this refer to the analysis of observations or simulations? If the latter the word "reveal" seems to be at conflict with the above statement. Furthermore, later the authors list many reasons for differences between the simulations of this study and Chiodo et al. As the aim is to "partly rebut the conclusions" of C19 it would be important to be very clear. Are the simulation results really different? Or is it just a different interpretation of similar results.

*We refer to the correlations of the observed NAO index and the solar cycle. We now are more specific in the first sentence:*
*"Even though correlations for observations as in Fig. 5c are similar to what has been shown in a recent study (cf. Fig. 1b in Chiodo et al., 2019), our conclusions are very different in the light of our model results."*
*We here mean that the simulation results are different as our experimental design is also quite different as elaborated (ensemble vs. single runs, more realistic CMIP6 forcing vs. idealized forcing, historical runs vs. constant GHG forcing). However, we explicitly do not rule out that similar results could be found in the C19 data if, e.g., monthly means instead of DJF means were analyzed. Since we do not have access to that data, we cannot verify nor reject this hypothesis.*

I'm also confused by the new statement "seemingly discrepant results could be due to the analysis of DJF means in most studies, which likely are not sensitive enough to capture the signal reliably". Does this refer to the papers cited in the bracket

above "(Gray et al. ...)"? Why do they show signals if they used the not sensitive enough DJF means? If this refers to other studies, please cite.

*We suggest to modify the paragraph:*

*However, as we show here, the solar signal is likely not present in all winter months. Therefore, in some studies there remains a signal in DJF means, while it could be averaged out in others, including ours (see below). This finding is also supported by the very recent study of Kuroda et al. (2022).*

*Furthermore, we adjust the text further down:*

*" We find statistically significant ensemble mean solar-cycle-induced surface signals in February during the strong epoch which are consistent with the top-down propagation from stratospheric signals. This limitation of the solar signal to February is also confirmed by Kuroda et al. (2022) using long observational datasets as well as a historical simulation with a different chemistry-climate model. A controversial  study (Chiodo et al., 2019) ..."*

L61 The authors contrast forced signals in the extratropical North Atlantic by saying that "up to 25% of decadal variance" is explained by the solar cycle and "this region shows low potential predictability due to other forcings. While this is not wrong it sounds like the solar influence is large compared to other forcings, which is not true. Averaging over this region by eye, I'd suggest that other forcings are still more important even in this region. It would be good to rephrase these sentences in order to avoid misinterpretation.

*The reviewer is correct that this sentence has not been clear enough. We rephrased it to:*

*"The extratropical North Atlantic is a hotspot of solar cycle influence on climate predictability (Fig. 1a). In some parts, up to 25% of the decadal variance of winter surface air temperatures are explained by the solar cycle in our model. At the same time, similar parts of the North Atlantic show comparably low (<15%) potential predictability due to other (low-frequency) external forcings (Fig. 1b) and large internal variability (>65%) (Fig. 1c)."*

*We hope that this clarifies that this statement refers to only parts of the North Atlantic, and that it is similar regions that have a relatively large solar PPVF, low PPVF due to other external forcings, and large internal variability. We also added the numbers to the text for an easier impression while reading.*

L80 "Consequently, solar variability and an adequate representation of its impact on climate is key to exploit the solar-induced potential predictability for decadal climate predictions." This sentence sounds odd. Of course solar variability is key to exploit the predictability potential created by itself, and how could it be done in models if the representation of its impact was inadequate.

*We understand the reviewer's irritation regarding this sentence. We have to admit that various revisions in the process of writing the manuscript resulted in a too short description. We wanted to stress here that the exploitation of solar-induced predictability by means of a climate model requires a proper solar forcing as well as a climate model that is capable of simulating the whole top-down mechanism. This may seem trivial in the context of our study but we consider it worth mentioning that most existing climate model simulations do not match these requirements. Only recently, standard GCMs as used for the CMIP exercise started to employ spectral solar irradiance as forcing. A number of CMIP6 models still use only total solar irradiance which leads to an underestimation of the UV variability, which is the basic element inducing the top-down mechanism. When it comes to decadal climate prediction, there is - to our knowledge - no single prediction system including a model component for stratospheric chemistry. However, the ozone forcing recommended for usage in CMIP6 is compiled as three-dimensional fields evolving in time, including variability induced by the solar variability. Compared to earlier CMIP exercises, CMIP6 models hence may be able to model some response to solar variability even if not employing interactive chemistry.*
*We try to express that by rephrasing the respective sentence:*

*"Consequently, decadal climate prediction systems might benefit from including realistic solar forcing (e.g. SSI instead of TSI) and an adequate representation of its impact on climate (by usage of interactive chemistry or at least an ozone forcing matching the solar variability), exploiting the solar-induced potential predictability."*

L117 "This shows that the response to the solar cycle is highly non-linear and not necessarily proportional to the forcing." Isn't "non-linear: and "not proportional" the same? Why then the very different characterizations ("highly" vs. "not necessarily"?). I would agree with the latter but not the first statement. To show that the response is non-linear one would need to show that the response of the strong epoch scaled to the weak-epoch forcing is statistically significant different from the weak-epoch response.

*We would like to refer to Figures S4 and S5, which show zonal mean temperature and wind during the weak and the strong epoch. In our opinion, it is clear from those plots that the response - compare weak and strong epoch - is not linearly scalable as suggested by the reviewer, i.e., the pattern is different, even reversed at times, and it is not the same pattern with reduced magnitude in the weak epoch as*

*compared to the strong epoch. However, we agree to align the nuances and suggest to delete "highly", while keeping the "not proportional" part as a clarification:*

*"This shows that the response to the solar cycle is  non-linear and not necessarily proportional to the forcing."*

L158 "the correlations reach statistical significances of 90% in the model, and in observations with a lag of two years " Although the different lags are mentioned in the preceding sentence, this sentence on its own can easily be misunderstood. The different-lag issue should be picked up again. Besides, I don't think that one more line (as kindly produce in the response to my earlier review would make the figure too busy. So please include it. Furthermore, The correlations reach barely outside the 90% range. And, leaving aside autocorrelation issues, wouldn't one expect 10% of the values to be outside this range accidentally?

1. *The figure is now updated.*
2. *True, one would expect 10% of the values to be outside accidentally. Note that there are some values "significant" 1890 and 1910 (for the model), which seems to be rather accidental.*
3. *As a side note and described in the Methods, we took into account autocorrelation of the time series.*
4. *We suggest the following changes of the sentences:*

*"Their running correlation for all overlapping 45-year windows is fluctuating in the earlier years but begins to rise in the 1920's both for the model (at 0 lag) and observations (first with a lag of 2 years, later 0 years) (Fig. 5c, cf. Fig. 7a in Kuroda et al., 2022, who also see a shift of the lag from +2 to 0 in observations around 1975 as shown here). In the second half of the 20th century, the correlations reach statistical significances of 90%  for the model NAO at 0 lag, while in observations significant correlations appear when the NAO lags the solar cycle by two years (Gray et al., 2013; Thiéblemont et al., 2015) and since the 1980's without lag"*

L163 "organization and synchronization of internal variability" Is organization different to synchronization? Please explain. And I'd say the NAO index is synchronized, internal variability remains internal variability.

*The reviewer is right that differences between "organization" and "synchronization" are marginal. Since we think that "synchronization" might imply a phase-locking to 0 lag, while "organization" does not, we prefer the term "organization" and delete "synchronization". We also agree that external forcing can only interfere with natural, and not internal, variability. Hence we now write "NAO":*

*"organization  of  the NAO"*

L193 The meaning of the "hence" is not clear to me.

*We deleted "hence".*

L207 "The NAO in turn is organized and synchronized by the solar cycle. "

Not sure why "in turn". Besides there is again the issue of organization and synchronization.

*We rephrased this sentence to:*

*"We show that a stronger solar cycle signal induces a surface response that resembles the NAO, and the NAO-  is organized  by the solar cycle."*